# HOTSPOT-DRIVEN PEPTIDE DESIGN VIA MULTI-FRAGMENT AUTOREGRESSIVE EXTENSION

**Jiahan Li**[1*]**, Tong Chen**[2*]**, Shitong Luo**[3]**, Chaoran Cheng**[4]**, Jiaqi Guan**[5]**, Ruihan Guo**[6]**,
Sheng Wang**[2]**, Ge Liu**[4]**, Jian Peng**[6]**, Jianzhu Ma**[1]
[1]Tsinghua University, [2]University of Washington, [3]Massachusetts Institute of Technology,
[4]University of Illinois Urbana-Champaign, [5]ByteDance Inc., [6]Helixon Inc.
ced3ljhypc@gmail.com  majianzhu@tsinghua.edu.cn

## ABSTRACT

Peptides, short chains of amino acids, interact with target proteins, making them a unique class of protein-based therapeutics for treating human diseases. Recently, deep generative models have shown great promise in peptide generation. However, several challenges remain in designing effective peptide binders. First, not all residues contribute equally to peptide-target interactions. Second, the generated peptides must adopt valid geometries due to the constraints of peptide bonds. Third, realistic tasks for peptide drug development are still lacking. To address these challenges, we introduce **PepHAR**, a hot-spot-driven autoregressive generative model for designing peptides targeting specific proteins. Building on the observation that certain hot spot residues have higher interaction potentials, we first use an energy-based density model to fit and sample these key residues. Next, to ensure proper peptide geometry, we autoregressively extend peptide fragments by estimating dihedral angles between residue frames. Finally, we apply an optimization process to iteratively refine fragment assembly, ensuring correct peptide structures. By combining hot spot sampling with fragment-based extension, our approach enables *de novo* peptide design tailored to a target protein and allows the incorporation of key hot spot residues into peptide scaffolds. Extensive experiments, including peptide design and peptide scaffold generation, demonstrate the strong potential of **PepHAR** in computational peptide binder design. The source code will be available at https://github.com/Ced3-han/PepHAR.

## 1 INTRODUCTION

Peptides, typically composed of 3 to 20 amino acid residues, are short single-chain proteins interacting with target proteins. (Bodanszky, 1988). Peptides play essential roles in various biological processes, including cellular signaling and immune responses (Petsalaki & Russell, 2008). They are emerging as a promising class of therapeutic drugs for complex diseases such as diabetes, obesity, hepatitis, and cancer (Kaspar & Reichert, 2013). Currently, there are approximately 80 peptide drugs on the global market, 150 in clinical development, and $400 - 600$ undergoing preclinical evaluation (Craik et al., 2013; Fosgerau & Hoffmann, 2015; Muttenthaler et al., 2021; Wang et al., 2022). Traditional peptide discovery methods rely on labor-intensive techniques like phage/yeast display for screening mutagenesis libraries (Boder & Wittrup, 1997; Wu et al., 2016), or energy-based computational tools to score candidate peptides (Raveh et al., 2011; Lee et al., 2018; Cao et al., 2022), both of which face limitations due to the immense combinatorial design space.

Recently, deep generative models, particularly diffusion and flow-based methods, have shown substantial promise in *de novo* protein design (Huang et al., 2016; Luo et al., 2022; Watson et al., 2022; Yim et al., 2023a; Bose et al., 2023). Given the compact relationship between the structure and sequence of peptides and their target proteins (Grathwohl & Wüthrich, 1976; Vanhee et al., 2011), a few methods have successfully designed peptides conditioned on target information (Xie et al., 2023; Li et al., 2024a; Lin et al., 2024). These approaches typically represent peptide residues as rigid frames in the SE(3) manifold, angles in a torus manifold, and types in a statistical manifold (Cheng et al., 2024; Davis et al., 2024). Encoder-decoder architectures, particularly flow-matching methods Lipman et al. (2022), are then used to generate all residues simultaneously.

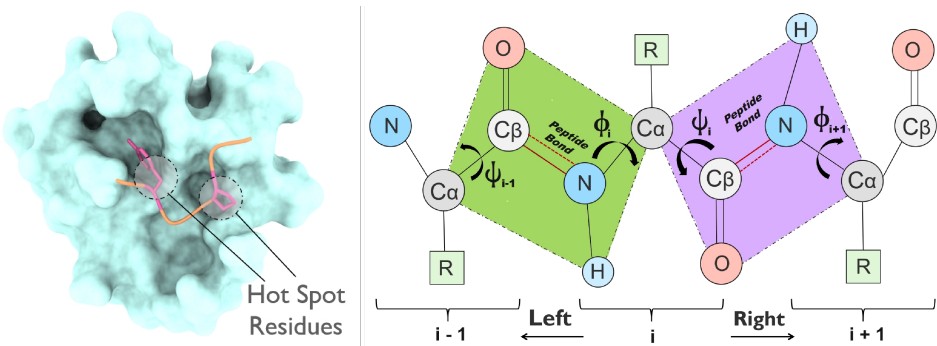

Figure 1: **Left**: Hot spot residues are a small number of critical residues in the peptide-target binding interface, while the remaining residues act as scaffolds. In peptide design, we first sample the hot spots and then use scaffold residues to link them. **Right**: Each residue in the protein consists of backbone heavy atoms and side-chain groups. Adjacent residues are connected by peptide bonds, which establish a planar conformation around neighboring atoms. The backbone structure of adjacent residues can be reconstructed using dihedral angles through the operations **Left** and **Right**.

Although these methods have initially succeeded in generating peptide binders with native-like structures and high affinities, several challenges remain. First, not all peptide residues contribute equally to binding. As shown in Figure 1, some residues establish key functional interactions with the target, possessing high stability and affinity. These are referred to as *hot spot residues* and are critical in drug discovery (Bogan & Thorn, 1998; Keskin et al., 2005; Moreira et al., 2007). Other residues, known as scaffolds, help position the hot spots in the binding region and stabilize the peptide (Matson & Stupp, 2012; Hosseinzadeh et al., 2021). Considering the different roles of these residues, generating all of them in one step may be inefficient. Second, the generated peptides must respect the non-rotatable constraints imposed by peptide bonds, which enforce fixed bond lengths and planar structures (Fisher, 2001). As illustrated in Figure 1, adjacent residues must maintain specific relative positions to form proper peptide bonds. A model that represents peptide backbone structures independently as local frames (Jumper et al., 2021) may neglect these geometric constraints. Third, peptides are not always designed from scratch in practical drug discovery. Initial peptide candidates are often optimized, or key hot spot residues are linked via scaffold residues (Zhang et al., 2009; Yu et al., 2023). Thus, more realistic in-silico benchmarks are needed to simulate these scenarios.

To tackle these challenges, we propose **PepHAR**, a hot-spot-driven autoregressive generative model. By distinguishing between hot spot and scaffold residues, we break down the generation process into three stages. First, we use an energy-based density model to capture the residue distribution around the target and apply Langevin dynamics to sample statistically favorable and feasible hot spots. Next, instead of generating all residues simultaneously, we autoregressively extend fragments step by step, modeling dihedral angles parameterized by a von Mises distribution to maintain peptide bond geometry. Finally, since the generated fragments may not align perfectly, we apply a hybrid optimization function to assemble the fragments into a complete peptide. To simulate practical peptide drug discovery scenarios, we evaluate our method not only in *de novo* peptide design but also in scaffold generation, where the model scaffolds known hot spot residues into a functional peptide, akin to peptide design based on prior knowledge.

In summary, our key contributions are:

- We introduce **PepHAR**, an autoregressive generative model based on hot spot residues for peptide binder design;

- We address current challenges in peptide design by using an **energy-based model** for hot spot identification, **autoregressive fragment** extension for maintaining peptide geometry, and an **optimization step** for fragment assembly;

- We propose a new experimental setting, **scaffold generation**, to mimic practical scenarios and demonstrate the competitive performance of our method in both peptide binder design and scaffold generation tasks.

## 2 RELATED WORK

**Generative Models for Protein Design** Generative models have shown significant promise in designing functional proteins (Yeh et al., 2023; Dauparas et al., 2023; Zhang et al., 2023c; Wang et al., 2021; Trippe et al., 2022; Yim et al., 2024). Some approaches focus on generating protein sequences using protein language models (Madani et al., 2020; Verkuil et al., 2022; Nijkamp et al., 2023) or through methods like directed evolution (Jain et al., 2022; Ren et al., 2022; Khan et al., 2022; Stanton et al., 2022). Others aim to design sequences based on backbone structures (Ingraham et al., 2019; Jing et al., 2020; Hsu et al., 2022; Li et al., 2022; Gao et al., 2022; Dauparas et al., 2022). For protein structures, which are crucial for determining protein function, diffusion-based (Luo et al., 2022; Watson et al., 2022; Yim et al., 2023b) and flow-based models (Yim et al., 2023a; Bose et al., 2023; Li et al., 2024a; Cheng et al., 2024) have been successfully applied to both unconditional (Campbell et al., 2024) and conditional protein design (Yim et al., 2024). However, these generative models typically treat all residues as equal, generating them simultaneously and overlooking the distinct roles of residues, such as those involved in catalytic sites (Giessel et al., 2022) or binding regions (Li et al., 2024a).

**Computational Peptide Design** The earliest methods for peptide design rely on protein or peptide templates for design (Bhardwaj et al., 2016; Hosseinzadeh et al., 2021; Swanson et al., 2022). These approaches use heuristic rules to search for similar sequences or structures in the PDB database as seeds for peptide design. A more prevalent class of models focuses on optimizing hand-crafted or statistical energy functions for peptide design (Cao et al., 2022; Bryant & Elofsson, 2023). While effective, these methods are computationally expensive and tend to get stuck in local minima (Raveh et al., 2011; Alford et al., 2017). Recently, deep generative models, such as GANs (Xie et al., 2023), diffusion models (Xie12 et al.; Wang et al., 2024), and flow models (Li et al., 2024a; Lin et al., 2024), have been applied to design peptide structures and sequences, conditioned on target protein information, offering more flexibility and efficiency in the design process.

## 3 PRELIMINARY

**Protein Composition** A protein or peptide is composed of multiple amino acid residues, each characterized by its type and backbone structure, which includes both position and orientation (Jumper et al., 2021). For the $i$-th residue, denoted as $R_i = (c_i, \boldsymbol{x}_i, \boldsymbol{O}_i)$, its type $c_i \in \{1...20\}$ refers to the class of its side-chain R group. The backbone position $\boldsymbol{x}_i \in \mathbb{R}^3$ represents the coordinates of the central C$\alpha$ atom, while the backbone orientation $\boldsymbol{O}_i \in \mathrm{SO}(3)$ is defined by the spatial configuration of the heavy backbone atoms (N-C$\alpha$-C). In this way, a protein can be represented as a sequence of $N$ residues: $[R_1, \ldots, R_N]$.

**Problem Formation** The goal of this work is to generate a peptide $D = [R_1, \ldots, R_N]$, consisting of $N$ residues, based on a target protein $T = [R_1, \ldots, R_M]$ of length $M$. We also define fragments, where the $k$-th fragment is denoted as $F_{(k,i_k,l_k)} = [R_{i_k}, \ldots, R_{i_k+l_k-1}]$, a contiguous subset of residues. Fragments are sequentially connected within the protein, where $i_k$ indicates the N-terminal residue's index of the fragment in the original peptide, and $l_k$ represents the fragment's length. Multiple fragments can be assembled into a complete protein based on their residue indices.

**Directional Relations** The sequential ordering from the N-terminal to the C-terminal residue, along with the covalent bonds between adjacent residues, is fundamental in our approach. As illustrated in Fig 1, residues are linked via covalent peptide bonds (CO-NH), with each residue $R_i$ connecting to its neighboring residues $R_{i-1}$ and $R_{i+1}$. These peptide bonds are partially double bonds, limiting their rotational freedom and resulting in a planar configuration for atoms between adjacent residues (C$_\alpha$, C$_\beta$, O, and H atoms). The backbone structure of a protein can thus be described using dihedral angles, which define the spatial relations between these planes in 3D space. Each residue has three associated dihedrals: $\psi_i$, $\phi_i$, and $\omega_i$. The first two angles determine the geometric relationship between adjacent residues, while the third controls the position of the O atom. Given a protein's backbone structure, we can calculate the dihedral angles for each residue. Conversely, the backbone structure of neighboring residues can also be derived from the dihedral angles, which serve as the building blocks in our model. Specifically, given the backbone position $\boldsymbol{x}_i$ and orientation $\boldsymbol{O}_i$, we can approximate the backbone structures of neighboring residues $R_{i-1}$ and $R_{i+1}$ using coordinate

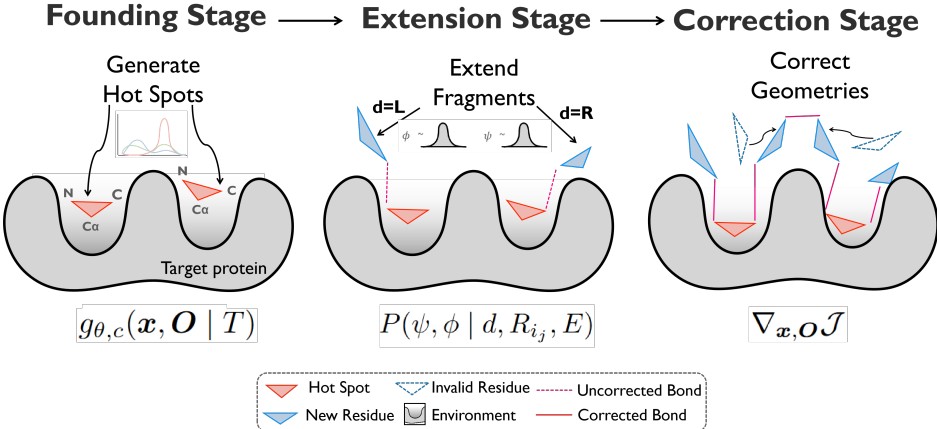

Figure 2: Overview of our three-stage approach: In the first foudning stage, $k$ hot-spot residues are generated ($k = 2$ in this example) from learned residue distribution around the target. New residues are extended to the fragments' left or right in the second stage based on dihedral angle distributions. Finally, in the correction stage, gradients from the objective functions are applied to refine the complete peptide.

transformations. Details are included in the Appendix B.

$$(\boldsymbol{x}_{i-1}, \boldsymbol{O}_{i-1}) = \textbf{Left}(\boldsymbol{x}_i, \boldsymbol{O}_i, \psi_{i-1}, \phi_i), \tag{1}$$

$$(\boldsymbol{x}_{i+1}, \boldsymbol{O}_{i+1}) = \textbf{Right}(\boldsymbol{x}_i, \boldsymbol{O}_i, \psi_i, \phi_{i+1}). \tag{2}$$

## 4 METHODS

To tackle the challenges in peptide design, we propose a three-stage approach to generate peptides $D$ based on their target protein $T$. Our method generates hot-spot residues, extends fragments, and corrects peptide structures. As shown in Figure 2 and Algorithm 1, the first stage, the **Founding Stage**, independently generates a small number $k$ of hot-spot residues $R_{i_1}, ..., R_{i_k}$ from the learned residue distribution $P(R \mid T)$. In the second stage, the **Extension Stage**, these hot-spot residues are used as starting points to progressively extend $k$ fragments $F_1, ..., F_k$ by adding new residues to the **Left** or **Right** in an autoregressive manner, until the total peptide reaches the desired length $L$. Finally, since each fragment is extended independently, the third stage, the **Correction Stage**, adjusts the sequences and structures of the fragments, refining them based on the gradients of the objective function to ensure valid geometries and meaningful peptide sequences.

### 4.1 FOUNDING STAGE

The founding stage first generates $k$ hot-spot residues based on the target protein $T$. By introducing the residue distribution $P(R \mid T)$, hot-spot residues represent those with higher probabilities (i.e., lower energies) of appearing near the binding pocket compared to other backbone structures and residue types. The generation of hot-spot residues focuses on finding regions with high probabilities, where residues are more likely to interact directly with the target. Conversely, regions with low probabilities (i.e., high energies) will have fewer peptide residues. For example, peptide residues should neither occur far from nor close to the pocket (Cao et al., 2022; Li et al., 2024a), as such regions may have near-zero probabilities (high energies) of residue occurrence. We parameterize $P(R \mid T)$ using an energy-based model, which is a conditional joint distribution of backbone position $\boldsymbol{x}$, orientation $\boldsymbol{O}$, and residue type $c$:

$$P_\theta(c, \boldsymbol{x}, \boldsymbol{O} \mid T) = \frac{1}{Z} \exp\left(g_{\theta,c}(\mathbf{x}, \mathbf{O} \mid T)\right). \tag{3}$$

here $g_\theta$ is a scoring function parameterized by an equivariant network, which quantifies the score of residue type $c$ occurring at a given backbone structure with position $\boldsymbol{x}$ and orientation $\boldsymbol{O}$ In other words, $g_{\theta,c}$ is the unnormalized probability of type $c$. And $Z$ is the normalizing constant associated with sequence and structure information, which we do not explicitly estimate.

---

**Algorithm 1:** Peptide Sampling Outline

---

**Data:** Target protein $T$, peptide length $N$, hot-spot residue count $k$, and indices $[i_1, ..., i_k]$

1  **Founding Stage**
2  **for** $j \leftarrow 1$ *to* $k$ **do**
3   | Sample a hot-spot residue $R_{i_j} \sim P_\theta(c, \boldsymbol{x}, \boldsymbol{O} \mid T)$ based on Eq. 6, 7, and 8;
4   | Initialize fragment $F_{(j, i_j, l_j=1)} \leftarrow \left[ R_{i_j} \right]$;
5  **Extension Stage**
6  **while** $l_1 + ... + l_k < N$ **do**
7   | Randomly choose a fragment index $i \in 1, ..., k$ and direction $d \in \{\text{L}, \text{R}\}$;
8   | Set the starting residue as either the N-terminal $R_{i_j}$ or the C-terminal $R_{i_{j+l_j-1}}$;
9   | Sample a new residue on the left $R_{i_j-1}$ or on the right $R_{i_{j+l_j}}$ based on Eq. 15 and 16;
10  | Add the new residue to fragment $F_j$;
11 Merge fragments into the peptide $D \leftarrow F_1 + ... + F_k$
12 **Correction Stage**
13 **for** $t \leftarrow 1, ...$ **do**
14  | Calculate the objective $\mathcal{J}$ based on the current peptide using Eq.22;
15  | Update the peptide using gradients from Eq.23 and 24;
16 **return** $D = [R_1, ..., R_N]$

---

**Network Implementation** The density model $g_\theta$ is parameterized by the Invariant Point Attention backbone network (Jumper et al., 2021; Luo et al., 2022; Yim et al., 2023b), which is SE(3) invariant. It takes positive residues (peptide residues) and negative residues (perturbed residues) along with the target protein as input, encoding them into hidden representations. A shallow Multi-Layer Perceptron (MLP) is then used to classify residue types for likelihood evaluation.

**Training** We use the Noise Contrastive Estimation (NCE) to train this parameterized energy-based model (Gutmann & Hyvärinen, 2010). NCE distinguishes between samples from the true data distribution (positive points) and samples from a noise distribution (negative points). The positive distribution corresponds to the ground truth residue distribution of the peptide over the target $(c, \boldsymbol{x}, \boldsymbol{O}) \sim p(R \mid T)$, while the negative samples are drawn from the disturbed distribution $(c_{\text{neg}}, \boldsymbol{x}^-, \boldsymbol{O}^-) \sim p(\tilde{R} \mid T)$ by adding large spatial noises to the ground truth positions and orientations, labeled as type $c_{\text{neg}}$. As positive and negative data are sampled equally, the NCE objective for a single positive data point is:

$$l(c, \boldsymbol{x}, \boldsymbol{O}, \mid T) = \log \frac{\exp g_{\theta,c}(\boldsymbol{x}, \boldsymbol{O} \mid T)}{\sum_{c'} \exp g_{\theta,c'}(\mathbf{x}, \mathbf{O} \mid T) + p(c_{neg}, \boldsymbol{x}, \boldsymbol{O} \mid T)}. \tag{4}$$

As a common practice (Gutmann & Hyvärinen, 2012), we fix the negative probability $p(c_{\text{neg}}, \boldsymbol{x}, \boldsymbol{O} \mid T)$ as a constant, simplifying the evaluation of log-likelihoods for negative samples. The final loss function is given by:

$$\mathcal{L}^{NCE} = -\mathbb{E}_+ \left[ l(c, \boldsymbol{x}, \boldsymbol{O} \mid T) \right] - \mathbb{E}_- \left[ l(c_{\text{neg}}, \boldsymbol{x}^-, \boldsymbol{O}^- \mid T) \right]. \tag{5}$$

**Sampling** In the founding stage, we sample $k$ hot-spot residues from the learned energy-based distribution, where $k$ is kept small relative to the peptide length (e.g., $k = 1 \sim 3$ in our experiments). Since hot-spots are assumed to be sparsely distributed along the peptide (Bogan & Thorn, 1998), we approximately sample them independently. For each hot-spot residue, we employ the Langevin MCMC Sampling algorithm (Welling & Teh, 2011), starting from an initial guessed position $\boldsymbol{x}^0$ and orientation $\boldsymbol{O}^0$, and iteratively update them using the following gradients:

$$\boldsymbol{x}^{t+1} \leftarrow \boldsymbol{x}^t + \frac{\epsilon^2}{2} \sum_{c'} \nabla_{\boldsymbol{x}} g_{\theta,c'}(\boldsymbol{x}^t, \boldsymbol{O}^t \mid T) + \epsilon \boldsymbol{z}_x^t, \boldsymbol{z}_x^t \sim \mathcal{N}(0, \text{I}_3), \tag{6}$$

$$\boldsymbol{O}^{t+1} \leftarrow \exp_{\boldsymbol{O}^t}(\frac{\epsilon^2}{2} \sum_{c'} \nabla_{\boldsymbol{O}} g_{\theta,c'}(\boldsymbol{x}^t, \boldsymbol{O}^t \mid T) + \epsilon \boldsymbol{Z}_O^t), \boldsymbol{Z}_O^t \sim \mathcal{TN}_{\boldsymbol{O}^t}(0, \text{I}_3), \tag{7}$$

$$c^{t+1} \sim \text{softmax} \, g_\theta(\boldsymbol{x}^t, \boldsymbol{O}^t \mid T). \tag{8}$$

Since orientation lies in the SO(3) space, we employ the exponential map and a Riemannian random walk on the tangent space for updates (De Bortoli et al., 2022). The summation of all possible

residue types ensures that we transition from regions of low occurrence probability to regions of higher probability. Finally, after each iteration, the residue type $c$ is sampled conditioned on the updated position and orientation.

## 4.2 EXTENSION STAGE

The extension stage expands fragments into longer sequences, starting from the sampled hot-spot residues. At each extension step, we add a new residue to either the left or right of fragment $F$. Based on the relationship between adjacent residues (Eq.1,2), the backbone structure of the new residue is inferred from its dihedral angles and the structure of the adjacent residue, which is either the N-terminal or C-terminal residue of the fragment. Specifically, when connecting a new residue to residue $R_{i_j}$ in the $j$ th fragment $F_j$, we model the dihedral angle distribution $P(\psi, \phi \mid d, R_{i_j}, E)$, where $d \in \{\mathbf{L}, \mathbf{R}\}$ indicates the extension direction and $E$ represents the surrounding residues, including target $T$ and other residues in the currently generated fragments.

$$P(\psi, \phi \mid d, R_{i_j}, E) = \begin{cases} P(\psi_{i_j-1}, \phi_{i_j}), & d = \mathbf{L}, \\ P(\psi_{i_j}, \phi_{i_j+1}), & d = \mathbf{R}. \end{cases} \tag{9}$$

Since multiple angles are involved, the dihedral angle distribution is modeled as a product of parameterized von Mises distributions (Lennox et al., 2009), which use cosine distance instead of L2 distance to measure the difference between angles, behaving like circular normal distributions. For example, when $d = \mathrm{L}$, we have:

$$P(\psi_{i_j-1}, \phi_{i_j}) = f_{\mathrm{VM}}(\psi; \mu_{\psi_{i_j-1}}, \kappa_{\psi_{i_j-1}}) f_{\mathrm{VM}}(\phi_{i_j}; \mu_{\phi_{i_j}}, \kappa_{\phi_{i_j}}), \tag{10}$$

$$f_{\mathrm{VM}}(\psi_{i_j-1}; \mu_{\psi_{i_j-1}}, \kappa_{\psi_{i_j-1}}) = \frac{1}{2\pi I_0(\kappa_{\psi_{i_j-1}})} \exp\left(\kappa_{\psi_{i_j-1}} \cdot cos(\mu_{\psi_{i_j-1}} - \psi_{i_j-1})\right), \tag{11}$$

$$f_{\mathrm{VM}}(\phi_{i_j}; \mu_{\phi_{i_j}}, \kappa_{\phi_{i_j}}) = \frac{1}{2\pi I_0(\kappa_{\phi_{i_j}})} \exp\left(\kappa_{\phi_{i_j}} \cdot cos(\mu_{\phi_{i_j}} - \phi_{i_j})\right). \tag{12}$$

Here, $I_0(\cdot)$ denotes the modified Bessel function of the first kind of order 0. The four distribution parameters are predicted by a neural network $h_\theta$, called the prediction network. Similarly, for $d = \mathrm{R}$, the network predicts another set of four parameters:

$$h_\theta(d, R_{i_j}, E) = \begin{cases} (\mu_{\psi_{i_j-1}}, \kappa_{\psi_{i_j-1}}, \mu_{\phi_{i_j}}, \kappa_{\phi_{i_j}}), & d = \mathbf{L}, \\ (\mu_{\psi_{i_j}}, \kappa_{\psi_{i_j}}, \mu_{\phi_{i_j+1}}, \kappa_{\phi_{i_j+1}}), & d = \mathbf{R}. \end{cases} \tag{13}$$

**Network Implementation** The prediction network $h_\theta$ uses the same IPA backbone to extract features. However, to avoid data leakage during training, we employ directional masks in the Attention module since the neighboring backbone structures are known and dihedral angles can be derived analytically. For instance, if the direction is **Left**, residues can only attend to their neighbors on the right during attention updates, and vice versa for **Right**.

**Training** We optimize the network parameters using Maximum Likelihood Estimation (MLE) over directions $d \sim \{\mathbf{L}, \mathbf{R}\}$ and peptides in the peptide-target complex dataset. The MLE objective is given by:

$$\mathcal{L}^{MLE} = -\mathbb{E}\left[\log P(\psi, \phi \mid d, R_{i_j}, E)\right]. \tag{14}$$

**Sampling** During the extension stage, we generate $k$ fragments corresponding to $k$ hot-spot residues from the founding stage. The extension process is iterative, where fragments are autoregressively extended until the total peptide length (the sum of fragment lengths) reaches a predefined value (e.g., the length of the native peptide). Consider a one-step extension of fragment $F$ in direction $d$. The starting residue $R_{i_j}$ depends on the direction: $d = \mathrm{L}$ implies adding a residue to the left of the fragment, making $R_{i_j}$ the N-terminal residue (first residue); $d = \mathrm{R}$ implies adding to the right, making $R_{i_j}$ the C-terminal residue (last residue). The other residues in the fragment and target form the environment $E$. We then sample the dihedral angles for the new residue in the chosen direction from the predicted distribution, using $h_\theta$. For example, when $d = \mathbf{L}$:

$$\psi_{i_j-1} \sim f_{\mathrm{VM}}(\psi_{i_j-1}; h_\theta(d = \mathrm{L}, R_{i_j}, E)), \tag{15}$$

$$\phi_{i_j} \sim f_{\mathrm{VM}}(\phi_{i_j}; h_\theta(d = \mathrm{R}, R_{i_j}, E)). \tag{16}$$

Next, the backbone structure of the newly added residue $R_{i_j-1}$ is reconstructed using the transformations in Eq 1. The residue type is then estimated by the density model $g_\theta$ used during the founding stage:

$$(\boldsymbol{x}_{i_j-1}, \boldsymbol{O}_{i_j-1}) = \textbf{Left}(\boldsymbol{x}_{i_j}, \boldsymbol{O}_{i_j}, \psi_{i_j-1}, \phi_{i_j}), \tag{17}$$

$$c_{i_j-1} \sim \text{softmax}\, g_\theta(\boldsymbol{x}_{i_j-1}, \boldsymbol{O}_{i_j-1} \mid E). \tag{18}$$

Finally, the process is repeated for another randomly selected fragment and direction.

### 4.3 CORRECTION STAGE

Although we autoregressively extended each fragment, the resulting fragments may not form a valid peptide with accurate geometry. For example, some fragments may not maintain proper distances between each other, leading to broken peptide bonds, while others may have incorrect dihedrals or residue types in relation to the whole peptide and the target protein. Some fragments may also exhibit atom clashes within the target protein. Inspired by traditional methods using hand-crafted energy functions (Alford et al., 2017), we introduce a correction stage as a post-processing step to refine the generated peptides. Rather than relying on empirical functions, we use the learned, network-parameterized distributions from the first two stages to regularize the peptides.

For a generated peptide $D = [(c_1, \boldsymbol{x}_1, \boldsymbol{O}_1), ..., (c_N, \boldsymbol{x}_N, \boldsymbol{O}_N)]$, the dihedrals of each residue are derived based on the backbone structures of adjacent residues. To ensure self-consistency between dihedrals and backbone structures, we use them to estimate new backbone structures and compare them with the original ones. The distance between these backbone structures reflects the validity of the generated peptide with respect to peptide bond properties and planarity. We define the distance between two residues' backbone structures separately for position and orientation and derive the backbone objective considering both directions:

$$d((\boldsymbol{x}_i, \boldsymbol{O}_i), (\boldsymbol{x}_j, \boldsymbol{O}_j)) = \|\boldsymbol{x}_i - \boldsymbol{x}_j\|^2 + \|\log(\boldsymbol{O}_i) - \log(\boldsymbol{O}_j)\|^2, \tag{19}$$

$$\mathcal{J}_{bb} = -\sum_{i=2}^{N} d(\textbf{Left}(\boldsymbol{x}_i, \boldsymbol{O}_i, \psi_{i-1}, \phi_i) - (\boldsymbol{x}_{i-1}, \boldsymbol{O}_{i-1})) - \sum_{i=1}^{N-1} d(\textbf{Right}(\boldsymbol{x}_i, \boldsymbol{O}_i, \psi_i, \phi_{i+1}) - (\boldsymbol{x}_{i+1}, \boldsymbol{O}_{i+1})). \tag{20}$$

Additionally, the dihedral angles must conform to the learned distribution $P(\psi, \phi)$ to ensure correct geometric relationships between neighboring residues. This leads to the dihedral objective, which is similar to Eq. 14. However, in Eq. 14, we optimize the network parameters to fit the angle distribution, whereas here, we keep the learned networks fixed and update the dihedrals instead:

$$\mathcal{J}_{ang} = -\sum_{i=2}^{N} \log P(\psi_{i-1}, \phi_i) - \sum_{i=1}^{N-1} \log P(\psi_i, \phi_{i+1}). \tag{21}$$

The final optimization objective is a weighted sum of the backbone and dihedral objectives. We iteratively update the peptide's backbone structures by taking gradients, similar to the founding stage, but we optimize the entire peptide at each timestep. The density model $g_\theta$ predicts the residue types at the end of each update step. Unlike the founding stage, where we started from random structures, the correction stage begins with the complete peptide.

$$\mathcal{J}_{corr} = \lambda_{bb}\mathcal{J}_{bb} + \lambda_{ang}\mathcal{J}_{ang}, \tag{22}$$

$$(\boldsymbol{x}_i^{t+1}, \boldsymbol{O}_i^{t+1}) \leftarrow \text{update}(\boldsymbol{x}_i^t, \boldsymbol{O}_i^t, \nabla_{\boldsymbol{x}_i}\mathcal{J}, \nabla_{\boldsymbol{O}_i}\mathcal{J}, ), \tag{23}$$

$$c^{t+1} \sim \text{softmax}(\boldsymbol{x}^t, \boldsymbol{O}^t \mid E). \tag{24}$$

## 5 EXPERIMENTS

**Overview** We evaluate PepHAR and several baseline methods on two main tasks: (1) Peptide Binder Design and (2) Peptide Scaffold Generation. In Peptide Design, we co-generate both the structure and sequence of peptides based on their binding pockets within the target protein. However, in real-world drug discovery, prior knowledge—such as key interaction residues at the binding interface or

initial peptides for optimization—is often available. Therefore, we introduce Peptide Scaffold Generation to assess how well models can scaffold and link these key residues into complete peptides, reflecting more practical applications. Details are included in the Appendix E.

**Dataset** Following Li et al. (2024a), we construct our training and test datasets. This moderate-length benchmark is derived from PepBDB (Wen et al., 2019) and Q-BioLip (Wei et al., 2024), with duplicates and low-quality entries removed. The binding pocket is defined as the residues in the target protein which has heavy atoms lying in the 10Åradius of any heavy atom in the peptide. The dataset consists of 158 complexes across 10 clusters from mmseqs2 (Steinegger & Söding, 2017), with an additional 8,207 non-homologous examples used for training and validation.

## 5.1 PEPTIDE BINDER DESIGN

In Peptide Binder Design, we co-generate peptide sequences and structures conditioned on the binding pockets of their target proteins. The models are provided with both the sequence and structure of the target protein pockets and tasked with generating bound-state peptides.

Table 1: Evaluation of methods in the peptide design task. $K = 1, 2, 3$ is the number of hot spots.

| | Valid % ↑ | RMSD Å ↓ | SSR % ↑ | BSR % ↑ | Stability % ↑ | Affinity % ↑ | Novelty % ↑ | Diversity % ↑ | Success % ↑ |
|---|---|---|---|---|---|---|---|---|---|
| RFDiffusion | **66.04** | 4.17 | 63.86 | 26.71 | **26.82** | 16.53 | 53.74 | 25.39 | 25.38 |
| ProteinGenerator | 65.88 | 4.35 | 29.15 | 24.62 | 23.84 | 13.47 | 52.39 | 22.57 | 24.43 |
| PepFlow | 40.27 | **2.07** | 83.46 | **86.89** | 18.15 | **21.37** | 50.26 | 20.23 | **27.96** |
| PepGLAD | 55.20 | 3.83 | 80.24 | 19.34 | 20.39 | 10.47 | 75.07 | 32.10 | 14.05 |
| PepHAR ($K = 1$) | 57.99 | 3.73 | 79.93 | 84.17 | 15.69 | 18.56 | **81.21** | **32.69** | 23.00 |
| PepHAR ($K = 2$) | 55.67 | 3.19 | 80.12 | 84.57 | 15.91 | 19.82 | 79.07 | 31.57 | 22.85 |
| PepHAR ($K = 3$) | 59.31 | 2.68 | **84.91** | 86.74 | 16.62 | 20.53 | 79.11 | 29.58 | 25.54 |

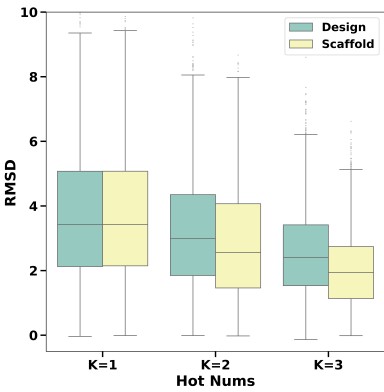

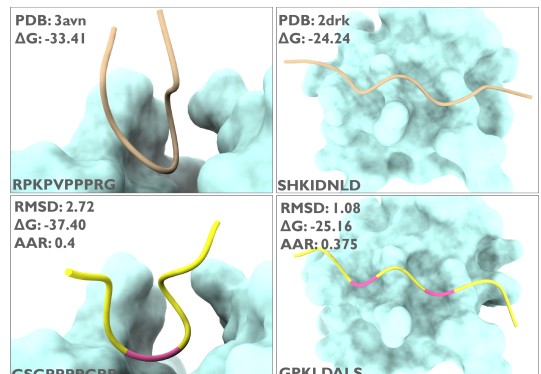

Figure 3: RMSD of generated peptides, considering different tasks and numbers of hotspots. More hotspot residues lead to better results.

Figure 4: Two examples of generated peptides, along with RMSD and binding energy. PepHAR can generate native-like peptides with better binding affinities.

**Metrics** A robust generative model should produce diverse, valid peptides with favorable stability and affinity. The following metrics are employed: (1) **Valid** measures whether the distance between adjacent residues is consistent with peptide bond formation, considering $C_\alpha$ atoms within $3.8\text{Å}$ as valid (Chelvanayagam et al., 1998; Zhang et al., 2012). (2) **RMSD** (Root-Mean-Square Deviation) compares the generated peptide structures to the native ones based on $C_\alpha$ distances after alignment. (3) **SSR** (Secondary Structure Ratio) evaluates the proportion of shared secondary structures between the generated and native peptides labeled by DSSP (Kabsch & Sander, 1983). (4) **BSR** (Binding Site Rate) assesses the similarity of peptide-target interactions by measuring the overlap of binding sites. (5) **Stability** calculates the percentage of generated complexes that are more stable (lower total energy) than their native counterparts, based on rosetta energy functions (Chaudhury et al., 2010; Alford et al., 2017). (6) **Affinity** measures the percentage of peptides with higher binding affinities (lower binding energies) than the native peptide. Beyond geometric and energetic factors, the model should also exhibit strong generalizability in discovering novel peptides. (7) **Novelty** is the ratio of novel peptides, defined by two criteria: (a) TM-score $\leq 0.5$ (Zhang & Skolnick, 2005) and (b) sequence identity $\leq 0.5$. (8) **Diversity** quantifies structural and sequence

variability, calculated as the product of pairwise (1-TM-score) and (1-sequence identity) across all generated peptides for a given target. (9) **Success** rate evaluates the proportion of AF2-predicted complex structures with a whole **ipTM** value higher than 0.6.

**Baselines** We compare PepHAR against three state-of-the-art peptide design models. RFDiffusion (Watson et al., 2022) uses pre-trained weights from RoseTTAFold (Baek et al., 2021) and generates protein backbone structures through a denoising diffusion process. Peptide sequences are then recovered using ProteinMPNN (Dauparas et al., 2022). ProteinGenerator augment RFDiffusion with sequence-structure jointly generation (Lisanza et al., 2023). PepFlow (Li et al., 2024a) models full-atom peptides and samples peptides using a flow-matching framework on a Riemannian manifold. PepGLAD (Kong et al., 2024) employs equivariant latent diffusion networks to generate full-atom peptide structures.

**Results** As shown in Table 1, PepHAR effectively generates peptides that exhibit valid geometries, native-like structures, and improved energies. While RFDiffusion produces valid peptides due to its pre-trained protein folding weights, PepFlow, which is trained solely on peptide datasets, struggles with generating valid peptides, making it challenging for practical applications. In contrast, PepHAR's autoregressive generation based on dihedral angles ensures the production of valid peptides and allows for precise placement at the binding site with accurate secondary structures. Similar to previous work (Li et al., 2024a), RFDiffusion excels at generating stable peptide-target structures, while PepHAR demonstrates competitive performance compared to PepFlow. Additionally, PepHAR shows impressive results in terms of novelty and diversity, highlighting its potential for exploring peptide distributions and designing a wide range of peptides for real-world applications. Figure 3 illustrates two examples of peptides generated by PepHAR, which closely resemble the structures and binding sites of native peptides while exhibiting low binding energies, indicating high affinities for the target.

## 5.2 PEPTIDE SCAFFOLD GENERATION

Compared to designing peptides from scratch, a more practical approach involves leveraging prior knowledge, such as key interaction residues. We introduce this as the task of scaffold generation, where certain hot spot residues in the peptide are fixed, and the model must generate a complete peptide by connecting these residues. In this context, the generated peptide should incorporate the hot spot residues in the correct positions, effectively scaffolding them. Hot spot residues are selected based on their higher potential for interacting with the target protein. To identify these, we first calculate the energy of each residue using an energy function (Alford et al., 2017), then manually select residues that are both energetically favorable and sparsely distributed along the peptide sequence. These selected residues are fixed as the condition for scaffold generation.

Table 2: Evaluation of methods in the scaffold generation task. $K = 1, 2, 3$ is the number of hot spots.

| | Valid % ↑ | RMSD Å ↓ | SSR % ↑ | BSR % ↑ | Stability % ↑ | Affinity % ↑ | Novelty % ↑ | Diversity % ↑ | Success % ↑ |
|---|---|---|---|---|---|---|---|---|---|
| RFDiffusion ($K = 3$) | **69.88** | 4.09 | 63.66 | 26.83 | 20.07 | 21.26 | 55.03 | 26.67 | 23.15 |
| ProteinGenerator ($K = 3$) | 68.52 | 3.95 | 65.86 | 24.17 | 20.40 | **22.80** | 50.73 | 20.82 | 20.42 |
| PepFlow ($K = 3$) | 42.68 | 2.45 | 81.00 | 82.76 | 11.17 | 18.27 | 50.93 | 16.97 | **24.54** |
| PepGLAD ($K = 3$) | 53.51 | 3.84 | 76.26 | 19.61 | 12.22 | 18.27 | 50.93 | **30.99** | 14.85 |
| PepHAR ($K = 1$) | 56.01 | 3.72 | 80.61 | 78.18 | 17.89 | 19.94 | **80.61** | 29.79 | 20.43 |
| PepHAR ($K = 2$) | 55.36 | 2.85 | 82.79 | 85.80 | 19.18 | 19.17 | 74.76 | 25.32 | 22.09 |
| PepHAR ($K = 3$) | 55.41 | **2.15** | 83.02 | 88.02 | 20.50 | 20.65 | 72.56 | 19.68 | 21.45 |

**Baselines and Metrics** We use the same baselines and metrics as in the Peptide Design task. Specifically, for RFDiffusion and Protein Generator, the known hot spot residues, along with the target, are provided as an additional condition. For PepFlow, we modify the ODE sampling process by initializing it with the ground truth hot spot residues and ensuring the model only modifies the remaining residues. In our method, we replace the sampled hot spot residues with the known ground truth residue.

**Results** As shown in Table 2, PepHAR demonstrates excellent performance in scaffolding hot spot residues into complete peptides. Given that the hot spot residues have functional binding capabilities. In contrast, the scaffold residues contribute primarily to structural integrity, the generated peptides are expected to possess valid, native structures with high stability. PepHAR successfully generates valid and native-like structures, ensuring that scaffold residues do not disrupt interactions between hot spot residues, achieving the best scores in SSR and BSR. Moreover, PepHAR achieves

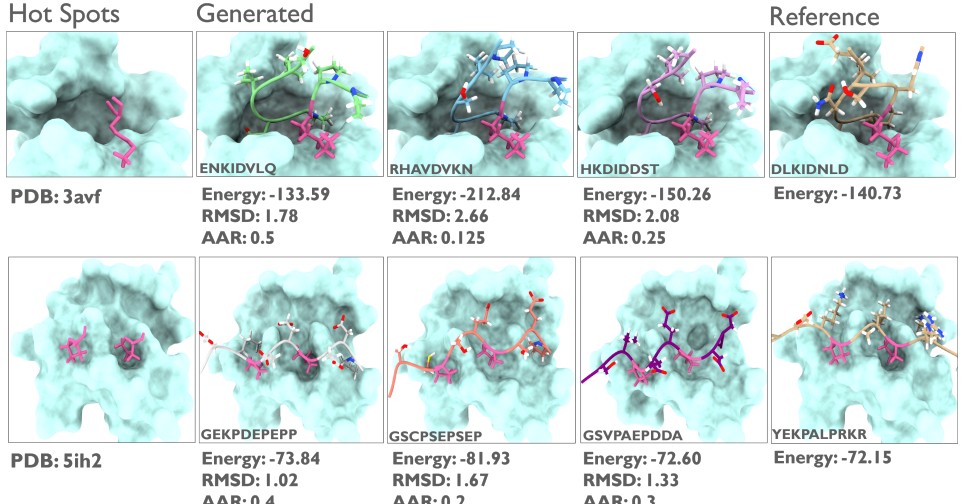

Figure 5: Examples of generated scaffolded peptides by PepHAR. PepHAR can scaffold hotspot residues, leading to more stable complexes with native-like valid geometries

competitive stability compared to RFDiffusion, which is trained on a larger PDB dataset. Additionally, PepHAR produces novel and diverse scaffolds. Figure 5 presents two examples of scaffolded peptides alongside native peptides and given residues. The generated scaffolds exhibit similar structures to the native ones, while displaying variations in geometry and orientation at the midpoints and ends of the peptides, indicating flexibility in the scaffolding regions. Furthermore, the generated scaffolds often have lower total energy than the native peptides, suggesting enhanced stability of the complex and improved interaction potential.

## 5.3 ANALYSIS

Table 3: Ablation results of PepHAR in peptide design task.

|  | Valid % ↑ | RMSD Å ↓ | SSR % ↑ | BSR % ↑ | Stability % ↑ | Affinity % ↑ | Novelty % ↑ | Diversity % ↑ |
|---|---|---|---|---|---|---|---|---|
| PepHAR ($K = 3$) | 59.31 | 2.68 | 84.91 | 86.74 | 16.62 | 20.53 | 79.11 | 29.58 |
| PepHAR w/o Von Mosies | 56.21 | 3.10 | 80.86 | 82.21 | 17.24 | 15.68 | 79.44 | 29.65 |
| PepHAR w/o Hot Spot | 55.67 | 3.99 | 79.93 | 74.17 | 11.23 | 12.21 | 81.51 | 37.03 |
| PepHAR w/o Correction | 53.66 | 3.41 | 80.46 | 81.43 | 15.72 | 14.85 | 82.75 | 37.87 |

**Effect of Hot Spots** Comparing Tables 1 and 2, we observe that introducing hot spots as prior knowledge significantly boosts PepHAR's performance while providing little benefit to RFDiffusion and PepFlow. This highlights PepHAR's versatility across different design tasks. We also investigate the effect of varying the number of hot spots, denoted as $K = 1, 2, 3$. As shown in Tables and Figure 3, increasing the number of hot spots improves geometries and energies, regardless of whether the hotspots are estimated by density models or provided as ground truth; however, it negatively impacts novelty and diversity. This illustrates a trade-off between designing low-diversity but high-quality peptides (in comparison to the native) and high-diversity but varied peptides (Luo et al., 2022; Li et al., 2024a).

**Ablation Study** Table 3 presents our ablation study, which assesses the effectiveness of different components in PepHAR. "PepHAR w/o Hot Spot" refers to the model where hot spots sampled from the density model are replaced with randomly positioned and typed residues. "PepHAR w/o Von Mises" indicates using direct angle predictions instead of modeling angle distributions. We also removed the correction stage from "PepHAR w/o Correction." Our findings reveal that generated hot spots are crucial for Valid, RMSD, SSR, and BSR metrics, underscoring their importance for achieving valid geometries and interactions. Modeling angle distributions also contributes positively by accounting for the flexibility of dihedral angles. Lastly, the final correction stage is vital in enhancing fragment assembly, leading to peptides with higher affinity and stability, which are essential for effective protein binding.

## 6 CONCLUSION

In this work, we presented **PepHAR**, a hot-spot-based autoregressive generative model designed for efficient and precise peptide design targeting specific proteins. By addressing key challenges in peptide design—such as the unequal contribution of residues, the geometric constraints imposed by peptide bonds, and the need for practical benchmarking scenarios—PepHAR provides a comprehensive approach for generating peptides from scratch or assembling peptides around key hot spot residues. Our method leverages energy-based hot spot sampling, autoregressive fragment extension through dihedral angles, and an optimization process to ensure valid peptide assembly. Through extensive experiments on both peptide generation and scaffold-based design, we demonstrated the effectiveness of PepHAR in computational peptide design, highlighting its potential for advancing drug discovery and therapeutic development.

## ACKNOWLEDGEMENTS

This work was supported by the National Key Plan for Scientific Research and Development of China (2023YFC3043300), China's Village Science and Technology City Key Technology funding, and Wuxi Research Institute of Applied Technologies, Tsinghua University under Grant 20242001120.

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

## A    HOT SPOT RESIDUES

In the context of protein interactions, hot spots refer to specific amino acid residues within a protein-protein interface that significantly contribute to the binding affinity and stability of the complex (Bogan & Thorn, 1998; Moreira et al., 2007; Keskin et al., 2005; Guo et al.). These residues are often characterized by their energetic contributions, with a few critical interactions having a disproportionate effect on the overall binding energy. These regions typically have a higher likelihood of binding events and are characterized by favorable structural and energetic features. Identifying and understanding these hot spots is crucial for the design of effective inhibitors or modulators that selectively target protein-protein interactions (PPIs). Several classic examples illustrate the concept of protein interaction hot spots:

- p53 and MDM2: The interaction between the tumor suppressor protein p53 and its negative regulator MDM2 is a well-studied example. Key residues in p53, such as Leu-22, Trp-23, and Phe-19, have been identified as hot spots for binding to MDM2. Disruption of this interaction is a promising strategy for cancer therapy (Vassilev, 2004).

- Antibody-Antigen Interactions: The binding of antibodies to their specific antigens often involves hot spots that are essential for the specificity and affinity of the interaction. For example, the interaction between the antibody 1G12 and the HIV-1 envelope glycoprotein gp120 features specific residues on both molecules that contribute significantly to binding (Zhou et al., 2007).

- Receptor-Peptide Interactions: The interaction between the CD4 receptor and the HIV-1 gp120 envelope protein is another notable example. Specific residues on CD4, such as Asp368 and Tyr371, serve as hot spots that facilitate binding, leading to viral entry into host cells (Sattentau & Moore, 1993).

- Cytokine-Receptor Binding: The interaction between cytokines and their receptors is critical for immune signaling. For instance, the binding of interleukin-6 (IL-6) to its receptor IL-6R involves hot spot residues that are crucial for signal transduction and biological activity (Rose-John et al., 2007).

In addition to hot-spot residues, which directly mediate protein-protein interactions, the remaining residues in the interface are referred to as scaffold residues. These scaffold residues play a crucial role in providing structural support to maintain the stability and conformation of the interface. While scaffold residues do not typically contribute significantly to the binding free energy, they ensure that the hot spots are properly positioned to interact with their binding partners. This structural framework is essential for maintaining the overall architecture of the protein complex and facilitating specific interactions at the hot-spot regions. For example, scaffold residues often help to shield hot spots from solvent exposure, thereby preserving their high binding affinities.

In our context, we focus on the hot-spot residues on the peptides that are crucial for key interactions with their target, as they are thought to be structurally conserved. While there are also hot spots on the target proteins, we do not model them here. Some works, however, focus on designing binders that specifically target these hot-spot residues on the proteins, treating them as critical interaction points (Watson et al., 2022; Zambaldi et al., 2024).

In our peptide design task, since we aim to design peptides from scratch, we lack prior knowledge about the ground-truth hot-spot residues, which are typically defined by energy functions and conserved interactions. Instead, we employ a density model to identify statistically favorable residues as hot spots. Essentially, our energy function is represented by a neural network. For the scaffold generation task, we first use the Rosetta energy function (Raveh et al., 2011; Alford et al., 2017) to compute the factorized binding energies of each peptide residue at the binding interface. We then manually select key residues with lower binding energy contributions, ensuring they are uniformly distributed along the peptide. This approach is important because, in scaffold generation, we aim to link these hot spots with a feasible and evenly distributed scaffold structure, ensuring the stability and functionality of each fragment.

# B  ADJACENT RESIDUE RECONSTRUCTION

## B.1  PEPTIDE BOND AND PLANAR

Peptide bonds are the key linkages between amino acids in proteins, formed through a condensation reaction between the carboxyl group of one amino acid and the amino group of another. This bond is an amide bond, specifically between the carbonyl carbon (C=O) of one amino acid and the nitrogen (N-H) of another. The nature of the peptide bond plays a crucial role in determining the structure and stability of peptides and proteins.

One significant characteristic of the peptide bond is its partial double bond nature. Although it's formally a single bond, resonance between the lone pair of electrons on the nitrogen and the carbonyl group gives the bond partial double-bond character. This resonance limits rotation around the peptide bond, which leads to a rigid and planar structure between the alpha carbon atoms of the amino acids involved in the bond. As a result, the peptide bond creates a flat, coplanar arrangement of the six atoms involved: the nitrogen, hydrogen, carbon, oxygen, and the two alpha carbons (one from each amino acid).

This planarity is crucial because it helps constrain the protein's overall folding pattern. The rigid planes of successive peptide bonds are connected by flexible single bonds at the alpha carbons, allowing the polypeptide chain to fold into its specific secondary structures like alpha helices and beta sheets. The restricted rotation around the peptide bond imposes dihedral angles, $\phi$ and $\psi$, which define the conformation of the polypeptide backbone and are critical in shaping the overall three-dimensional structure of proteins.

In summary, the partial double-bond character of the peptide bond is a key factor in maintaining the planarity and rigidity of the peptide backbone, which in turn plays a fundamental role in determining the folding and function of proteins.

## B.2  DIHEDRAL ANGLES

In proteins, the dihedral angles—$\phi$ and $\psi$—define the conformation of the protein backbone by describing the rotation around specific bonds in the peptide chain. These angles are crucial for understanding the three-dimensional structure of a protein. Below is a detailed explanation of how these angles are calculated:

**General Formula for Dihedral Angle Calculation**  To calculate a dihedral angle between four consecutive atoms $(A, B, C, D)$, the steps are:

1. Compute the bond vectors:

$$\vec{AB} = B - A, \quad \vec{BC} = C - B, \quad \vec{CD} = D - C$$

2. Calculate the normal vectors of the two planes formed by the atoms:

$$\vec{n}_1 = \vec{AB} \times \vec{BC}, \quad \vec{n}_2 = \vec{BC} \times \vec{CD}$$

3. Use the dot product to find the angle between the two planes:

$$\text{angle} = \arctan 2 \left( \vec{BC} \cdot (\vec{n}_1 \times \vec{n}_2), \, \vec{n}_1 \cdot \vec{n}_2 \right) \tag{25}$$

This general method can be applied to calculate $\phi$ and $\psi$ dihedral angles across a protein's backbone.

**Phi ($\phi_i$) Angle**  The $\phi_i$ angle is the dihedral angle around the bond between the nitrogen (N) and alpha carbon ($C_\alpha$) of residue $i$. It is defined by the following four atoms:

- $C_{i-1}$: Carbonyl carbon of the previous residue
- $N_i$: Amide nitrogen of the current residue
- $C_{\alpha,i}$: Alpha carbon of the current residue
- $C_i$: Carbonyl carbon of the current residue

The dihedral angle $\phi_i$ is calculated as the angle between the planes formed by the atoms $(C_{i-1}, N_i, C_{\alpha,i})$ and $(N_i, C_{\alpha,i}, C_i)$:

$$\phi_i = \text{angle between planes } (C_{i-1}, N_i, C_{\alpha,i}) \text{ and } (N_i, C_{\alpha,i}, C_i) \tag{26}$$

**Psi ($\psi_i$) Angle** The $\psi_i$ angle describes the dihedral angle around the bond between the alpha carbon ($C_\alpha$) and the carbonyl carbon (C) of residue $i$. It is defined by the following four atoms:

- $N_i$: Amide nitrogen of the current residue
- $C_{\alpha,i}$: Alpha carbon of the current residue
- $C_i$: Carbonyl carbon of the current residue
- $N_{i+1}$: Amide nitrogen of the next residue

The dihedral angle $\psi_i$ is calculated as the angle between the planes formed by the atoms $(N_i, C_{\alpha,i}, C_i)$ and $(C_{\alpha,i}, C_i, N_{i+1})$:

$$\psi_i = \text{angle between planes } (N_i, C_{\alpha,i}, C_i) \text{ and } (C_{\alpha,i}, C_i, N_{i+1}) \tag{27}$$

**Psi ($\psi_{i-1}$) of Previous Residue** The $\psi_{i-1}$ angle is calculated similarly to $\psi_i$ but for the previous residue. It involves the following four atoms:

- $N_{i-1}$: Amide nitrogen of the previous residue
- $C_{\alpha,i-1}$: Alpha carbon of the previous residue
- $C_{i-1}$: Carbonyl carbon of the current residue
- $N_i$: Amide nitrogen of the current residue

The dihedral angle $\psi_{i-1}$ is calculated as:

$$\psi_{i-1} = \text{angle between planes } (N_{i-1}, C_{\alpha,i-1}, C_{i-1}) \text{ and } (C_{\alpha,i-1}, C_{i-1}, N_i) \tag{28}$$

**Phi ($\phi_{i+1}$) of Next Residue** The $\phi_{i+1}$ angle is similar to $\phi_i$ but for the next residue. It involves the following four atoms:

- $C_i$: Carbonyl carbon of the current residue
- $N_{i+1}$: Amide nitrogen of the next residue
- $C_{\alpha,i+1}$: Alpha carbon of the next residue
- $C_{i+1}$: Carbonyl carbon of the next residue

The dihedral angle $\phi_{i+1}$ is calculated as:

$$\phi_{i+1} = \text{angle between planes } (C_i, N_{i+1}, C_{\alpha,i+1}) \text{ and } (N_{i+1}, C_{\alpha,i+1}, C_{i+1}) \tag{29}$$

## C  ADJACENT STRUCTURE RECONSTRUCTION

Reconstructing the backbone structures of adjacent residues $R_{i-1}$ and $R_{i+1}$ involves two main steps. First, we use dihedral angles to rotate the standard residue coordinates in the local frame. Then, we transform the local frame coordinates back to the global frame based on the position and orientation of the given residue. As an example, let's consider the **Right** operation, where we use the coordinates of $R_i$ and the associated dihedral angles $\psi_i$ and $\phi_{i+1}$ to compute the structure of $R_{i+1}$.

### C.1  CALCULATING $\mathbf{x}_i$ AND $\mathbf{O}_i$

To convert local coordinates into global coordinates, we first need to calculate the translation vector $\mathbf{x}_i$ and the orientation matrix $\mathbf{O}_i$ for residue $R_i$.

**Translation Vector $\mathbf{x}_i$**    The translation vector $\mathbf{x}_i$ defines the global position of residue $R_i$, and it is typically chosen as the position of a key atom within the residue. A common choice is the $\alpha$-carbon ($\mathbf{CA}_i$), which is centrally located in the backbone of the residue, providing a stable reference point for the rest of the structure.

$$\mathbf{x}_i = \mathbf{CA}_i \tag{30}$$

This ensures that the relative coordinates of other atoms in $R_i$ can be translated into the global coordinate system.

**Orientation Matrix $\mathbf{O}_i$**    The orientation matrix $\mathbf{O}_i$ defines the local frame of reference for residue $R_i$ and allows us to rotate local coordinates into the global frame. To construct $\mathbf{O}_i$, we use the positions of three key atoms: $\mathbf{C}_i$ (carbonyl carbon), $\mathbf{CA}_i$ (alpha carbon), and $\mathbf{N}_i$ (amide nitrogen). These atoms form the backbone of the residue, and their relative positions define the orientation of the local coordinate system.

The steps to compute $\mathbf{O}_i$ are as follows:

1. Compute the vector from $\mathbf{C}_i$ to $\mathbf{CA}_i$, and normalize it to obtain the first basis vector $\mathbf{e}_1$:

$$\mathbf{v}_1 = \mathbf{C}_i - \mathbf{CA}_i \tag{31}$$

$$\mathbf{e}_1 = \frac{\mathbf{v}_1}{||\mathbf{v}_1||} \tag{32}$$

2. Compute the vector from $\mathbf{N}_i$ to $\mathbf{CA}_i$, and remove the projection of this vector onto $\mathbf{e}_1$ to get the component orthogonal to $\mathbf{e}_1$. Normalize this to obtain the second basis vector $\mathbf{e}_2$:

$$\mathbf{v}_2 = \mathbf{N}_i - \mathbf{CA}_i \tag{33}$$

$$\mathbf{u}_2 = \mathbf{v}_2 - \left( \frac{\mathbf{v}_2 \cdot \mathbf{e}_1}{||\mathbf{e}_1||^2} \right) \mathbf{e}_1 \tag{34}$$

$$\mathbf{e}_2 = \frac{\mathbf{u}_2}{||\mathbf{u}_2||} \tag{35}$$

3. Compute the third orthogonal vector $\mathbf{e}_3$ as the cross product of $\mathbf{e}_1$ and $\mathbf{e}_2$:

$$\mathbf{e}_3 = \mathbf{e}_1 \times \mathbf{e}_2 \tag{36}$$

4. The orientation matrix $\mathbf{O}_i$ is formed by combining these three orthonormal vectors into a matrix:

$$\mathbf{O}_i = [\mathbf{e}_1, \mathbf{e}_2, \mathbf{e}_3] \tag{37}$$

This orientation matrix $\mathbf{O}_i$ defines the local coordinate system for residue $R_i$ and can be used to transform the local coordinates of neighboring residues into the global frame.

C.2    LOCAL COORDINATE RECONSTRUCTION

Given a reference peptide structure, we apply dihedral angle transformations to compute the new coordinates based on the angles $\psi_i$ and $\phi_{i+1}$.

**Rotation Matrix Definition**    The rotation matrix for an arbitrary axis $\mathbf{d} = (d_x, d_y, d_z)$ and angle $\theta$ is given by the Rodrigues' rotation formula:

$$R(\mathbf{d}, \theta) = I + \sin(\theta)\mathbf{K} + (1 - \cos(\theta))\mathbf{K}^2 \tag{38}$$

where $\mathbf{K}$ is the skew-symmetric matrix of $\mathbf{d}$:

$$\mathbf{K} = \begin{bmatrix} 0 & -d_z & d_y \\ d_z & 0 & -d_x \\ -d_y & d_x & 0 \end{bmatrix} \tag{39}$$

**Initial Reference Coordinates**  The initial reference coordinates for the peptide are as follows:

$$\mathbf{N1} = (-0.572, 1.337, 0.000), \quad \mathbf{CA1} = (0.000, 0.000, 0.000), \quad \mathbf{C1} = (1.517, 0.000, 0.000)$$

$$\mathbf{N2} = (2.1114, 1.1887, 0.0000), \quad \mathbf{CA2} = (3.5606, 1.3099, 0.0000), \quad \mathbf{C2} = (4.0913, -0.1112, 0.0000)$$

*Note: The above coordinates are derived from the standard structure of glycine (GLY), one of the simplest amino acids. GLY is chosen because it lacks a side chain (only a hydrogen atom as a side chain), making it an ideal reference for constructing peptide backbone geometries. Additionally, the peptide bond between* **C1** *and* **N2** *is assumed to be planar, with* $\psi_1 = \psi_2 = 0°$, *reflecting the typical rigidity of peptide bonds. This simplifies the geometric model by aligning the peptide bond in a 0/180-degree plane.*

**Rotate C2 around CA2 $\to$ N2 axis ($\phi_{i+1}$)**  We first rotate **C2** around the axis from **CA2** to **N2** by the dihedral angle $\phi_{i+1}$. The axis is defined as:

$$\mathbf{d}_{N2-CA2} = \frac{\mathbf{N2} - \mathbf{CA2}}{||\mathbf{N2} - \mathbf{CA2}||} \tag{40}$$

The new position of **C2** after applying the rotation matrix $R(\phi_{i+1})$ is:

$$\mathbf{C2}' = \mathbf{CA2} + R(\mathbf{d}_{N2-CA2}, \phi_{i+1}) \cdot (\mathbf{C2} - \mathbf{CA2}) \tag{41}$$

**Rotate C2 around CA1 $\to$ C1 axis ($\psi_i$)**  Next, we rotate **C2** around the axis from **CA1** to **C1** by the dihedral angle $\psi_i$. The axis is defined as:

$$\mathbf{d}_{C1-CA1} = \frac{\mathbf{C1} - \mathbf{CA1}}{||\mathbf{C1} - \mathbf{CA1}||} \tag{42}$$

The new positions of **C2**, **CA2**, and **N2** after applying the rotation matrix $R(\psi_i)$ are:

$$\mathbf{C2}_{\text{rel}} = \mathbf{CA1} + R(\mathbf{d}_{C1-CA1}, \psi_i) \cdot (\mathbf{C2}' - \mathbf{CA1}) \tag{43}$$

$$\mathbf{CA2}_{\text{rel}} = \mathbf{CA1} + R(\mathbf{d}_{C1-CA1}, \psi_i) \cdot (\mathbf{CA2} - \mathbf{CA1}) \tag{44}$$

$$\mathbf{N2}_{\text{rel}} = \mathbf{CA1} + R(\mathbf{d}_{C1-CA1}, \psi_i) \cdot (\mathbf{N2} - \mathbf{CA1}) \tag{45}$$

### C.3 Converting Relative Coordinates to Global Coordinates

Once the relative coordinates have been calculated, we transform them back into global coordinates using a combination of rotation $\mathbf{O}_i$ and translation $\mathbf{x}_i$. We use the calculated relative coordinates of $CA2$, $C2$, and $N2$ and transform them into the global coordinate system.

This transformation is applied to each of the atoms $CA2$, $C2$, and $N2$:

$$\mathbf{CA_{i+1}} = \mathbf{CA2}_{\text{global}} = \mathbf{x}_i + \mathbf{O}_i \cdot \mathbf{CA2}_{\text{rel}}, \tag{46}$$

$$\mathbf{C_{i+1}} = \mathbf{C2}_{\text{global}} = \mathbf{x}_i + \mathbf{O}_i \cdot \mathbf{C2}_{\text{rel}}, \tag{47}$$

$$\mathbf{N_{i+1}} = \mathbf{N2}_{\text{global}} = \mathbf{x}_i + \mathbf{O}_i \cdot \mathbf{N2}_{\text{rel}}. \tag{48}$$

After getting backbone atoms, the side-chain atoms are reconstructed by side-chain packing algorithms, in our implementation, we use *PackRotamersMover* in Pyrosetta (Chaudhury et al., 2010) to pack side-chains for the newly added residue, which is based on rotamer library and rosetta energy function (Shapovalov & Dunbrack, 2011; Alford et al., 2017).

## D  Von Mosies Distribution

The von Mises distribution is a continuous probability distribution defined on the circle, commonly used to model angular or directional data. It is sometimes referred to as the "circular normal distribution" due to its similarity to the normal distribution on a plane, but it is defined for angles or directions. The von Mises distribution is often referred to as the circular normal distribution because it shares several properties with the normal distribution, such as being unimodal and symmetric around the mean. As $\kappa$ approaches infinity, the von Mises distribution approaches a normal distribution in terms of angular deviation.

### D.1 PROBABILITY DENSITY FUNCTION (PDF)

The probability density function (PDF) of the von Mises distribution is given by:

$$f(\theta \mid \mu, \kappa) = \frac{1}{2\pi I_0(\kappa)} \exp\left(\kappa \cos(\theta - \mu)\right)$$

where:

- $\theta \in [0, 2\pi)$ is the random variable ((an angle measured in radians),

- $\mu$ is the mean direction of the distribution, or the central angle around which the data are clustered,

- $\kappa \geq 0$ is the concentration parameter,analogous to the inverse of the variance in the normal distribution, which controls how tightly the data are clustered around the mean direction. When $\kappa = 0$, the distribution is uniform around the circle, while larger values of $\kappa$ indicate that the data are more tightly concentrated around $\mu$,

- $I_0(\kappa)$ is the modified Bessel function of the first kind of order 0, which serves as a normalization constant to ensure that the total probability integrates to 1 over the circle, defined as:

$$I_0(\kappa) = \frac{1}{\pi} \int_0^{\pi} e^{\kappa \cos(\phi)} d\phi$$

The concentration parameter $\kappa$ controls the spread of the distribution around the mean direction $\mu$:

- When $\kappa = 0$, the von Mises distribution becomes a uniform distribution on the circle.

- As $\kappa \to \infty$, the distribution becomes increasingly concentrated around $\mu$ and approaches a normal distribution for small angular deviations.

### D.2 CUMULATIVE DISTRIBUTION FUNCTION (CDF)

The cumulative distribution function (CDF) of the von Mises distribution does not have a simple closed-form expression. However, it can be computed numerically as:

$$F(\theta \mid \mu, \kappa) = \frac{1}{2\pi I_0(\kappa)} \int_{-\pi}^{\theta} \exp\left(\kappa \cos(t - \mu)\right) dt$$

where the integral is typically evaluated numerically due to the complexity introduced by the Bessel function and the circular nature of the distribution.

### D.3 MEAN AND VARIANCE

The mean direction $\mu$ is the central tendency of the von Mises distribution, and the concentration parameter $\kappa$ influences how closely the data are clustered around $\mu$. The variance of the distribution is related to $\kappa$ as follows:

$$\text{Var}(\theta) = 1 - \frac{I_1(\kappa)}{I_0(\kappa)}$$

where $I_1(\kappa)$ is the modified Bessel function of the first kind of order 1.

In our implementation, we use VonMises distribution implemented in pytorch package for sampling angles and evluating likelihood.

# E PepHAR Implementations

## E.1 Network Details

The network used in our method includes the density model $g_\theta$ in the founding stage and the prediction model $h_\theta$ in the extension stage. Both networks use an Invariant Point Attention (IPA)-based encoder to encode hidden representations of peptide residues (Yim et al., 2023a; Luo et al., 2022; Li et al., 2024a; Wu et al., 2024b). The input to the network consists of the backbone coordinates and residue types of the peptide-target complex. Several shallow MLPs (multi-layer perceptrons) are applied to transform these inputs into initial feature representations. The network outputs node embeddings and node-pair embeddings.

For the node (residue) embeddings, we use a combination of the following features:

- Residue type: A learnable embedding is applied.

- Atom coordinates: This includes both backbone and side-chain atoms.

These features are processed by separate MLPs. The concatenated features are then transformed by another MLP to produce the final node embedding.

For the edge (residue-pair) embeddings, we use a combination of the following features:

- Residue-type pair: A learnable embedding matrix of size $20 \times 20$ is applied.

- Relative sequential positions: A learnable embedding of the relative position between the two residues is applied.

- Distance between two residues.

- Relative orientation between two residues: The inter-residue backbone dihedral angles are calculated to represent the relative orientation, and sinusoidal embeddings are applied.

Similarly, these features are processed by separate MLPs, concatenated, and then transformed by another MLP to produce the final edge embedding.

Starting from the node and edge embeddings, the IPA encoder further encodes these representations. The density model then applies a classification layer to classify residue types, while the prediction model uses a regression head to predict parameters of the angle distributions. Since we use two separate models instead of a single model, we set the embedding size to $128$ for node embeddings and $16$ for edge embeddings, ensuring comparable parameters to our baseline models (Li et al., 2024a). The IPA encoder consists of $4$ layers, each with $8$ query heads and a hidden dimension of $32$.

## E.2 Training Details

Both the density and prediction models are trained on 8 NVIDIA A100 GPUs. As we found these two models prone to overfitting, we employed an early stopping strategy, training the density model for $1400$ iterations and the prediction model for $2400$ iterations. This results in a training time that is significantly shorter than the baseline models. We set the batch size to 64, using Adam as the optimizer with a learning rate of $3 \times 10^{-4}$. To prevent overfitting, we also applied a dropout rate of $0.5$ at each layer in the IPA.

## E.3 Sampling Details

For the sampling process, we use 10 iterations with an update rate of $0.01$ in the founding stage to sample anchors by default, followed by $100$ fine-tuning steps with an update rate of $0.1$ in the correction stage. In the scaffold design task, we replace the sampled hotspot residues with predefined ground truth residues.

## F    EXPERIMENTAL DETAILS

In each task, for every method, we generate $64$ peptides for each target protein, and the evaluation metrics are averaged across all generated peptides.

### F.1    METRIC DETAILS

**Valid**    This metric checks whether the distance between adjacent residues is consistent with peptide bond formation. Specifically, the distance between the $C_\alpha$ atoms of adjacent residues must be within $3.8\mathring{A}$ for the peptide to be considered valid (Chelvanayagam et al., 1998; Zhang et al., 2012).

**RMSD**    The Root-Mean-Square Deviation (RMSD) is a widely used metric to assess structural similarity between two protein conformations. In our evaluation, the generated peptide is aligned with the native peptide using the Kabsch algorithm Kabsch (1976), focusing only on the peptide portion of the complex for superposition. After alignment, we compute the RMSD by calculating the normalized distance between corresponding $C_\alpha$ atoms in the generated and native peptides. A lower RMSD value indicates a closer alignment to the native structure, with 0 implying perfect alignment.

**SSR**    The Secondary Structure Ratio (SSR) measures the similarity between the secondary structures of the generated peptide and the reference peptide. It is computed by comparing the secondary structure labels of the two peptides. These labels are assigned using the DSSP software Kabsch & Sander (1983). The ratio of matching secondary structure assignments between the generated and native peptides is then calculated. A higher SSR value indicates that the generated peptide preserves the secondary structure of the native peptide more closely.

**BSR**    Binding Site Rate (BSR) quantifies how well the binding interactions of the generated peptide with the target protein resemble those of the native peptide. We define a residue as part of the binding site if its $C_\beta$ atom is within a 6Åradius of any atom in the peptide. The BSR is calculated as the overlap ratio of binding site residues between the generated peptide and the native peptide. A higher BSR indicates that the generated peptide interacts with the protein target in a manner similar to the native peptide, which could suggest similar functional properties.

**Stability**    Stability refers to the proportion of designed peptides that achieve a lower energy score than the native peptide-protein complex. A lower energy score generally implies greater structural stability. We use the *FastRelax* protocol in PyRosetta Chaudhury et al. (2010) to relax each complex, followed by evaluation using the *REF2015* scoring function. It is then computed as the fraction of complexes where the designed peptide leads to a lower total energy compared to the native complex, indicating improved stability.

**Affinity**    Binding Affinity evaluates the percentage of designed peptides that exhibit stronger binding interactions with the target protein compared to the native peptide, as determined by their binding energy. Higher binding affinity usually suggests enhanced peptide functionality. Using PyRosetta's *InterfaceAnalyzerMover* Chaudhury et al. (2010), we calculate the binding energy after relaxing the complex and defining the interaction interface. Affinity is the percentage of peptides that show lower binding energy (and thus higher affinity) relative to the native peptide.

**Novelty**    Novelty measures the proportion of generated peptides that are structurally and sequentially distinct from the native peptide. A peptide is considered novel if it satisfies two conditions: (a) the TM-score (a structural similarity score) between the generated and native peptides is less than or equal to 0.5 (Zhang & Skolnick, 2005), and (b) the sequence identity between the two peptides is less than or equal to 0.5. A higher novelty score indicates that more generated peptides are different from the native peptide.

**Diversity**    Diversity assesses the variability among the generated peptides in terms of both structure and sequence. It is calculated as the product of pairwise (1 - TM-score) and (1 - sequence identity)

across all generated peptides for a given target. A higher diversity score indicates greater structural and sequence diversity among the generated peptides, suggesting a broader range of potential functional variations.

**Success** Success rate evaluates the quality of predicted complex structures using the confidence score from AlphaFold2. Specifically, we employ AlphaFold2 Multimer (Evans et al., 2021) to predict the structures of peptides and their full-length receptors. The success rate is calculated as the proportion of complexes with an overall ipTM score greater than 0.6.

## F.2 BASELINE DETAILS

In the scaffold geneartion task, we set all baseline hot spot residues as $K = 3$, as the highest hot spot numbers in our experiment.

**RFDiffusion** For RFDiffusion (Watson et al., 2022), we use the official implementation along with the pretrained weights for the complex version. Specifically, we apply 200 discrete timesteps during the diffusion process, generating 64 peptides for each target protein. In the peptide design task, we use the official binder design scripts for sampling. For the scaffold design task, we provide the ground-truth hotspot residues as additional information, instructing the model to inpaint the remaining peptide structure. After sampling the peptide backbone, we use ProteinMPNN (Dauparas et al., 2022) to generate the corresponding peptide sequences.

**ProteinGenerator** ProteinGenerator jointly samples protein backbone structures and sequences (Lisanza et al., 2023). We utilize the official inference scripts, employing 200 diffusion timesteps to design 64 peptides for each target protein.

**PepFlow** We employ the released Model No.1 from PepFlow (Li et al., 2024a), which is a full-atom peptide design model based on multi-modal flow matching. In the peptide design task, we utilize the official implementation for sampling. For the scaffold generation task, we modify the sampling process to ensure that the hotspot residues are constrained to their ground-truth values during each update step. For each generation, we perform 200 discrete time steps for sampling 64 peptides of each target.

**PepGLAD** We utilize the released inference script and co-design model from PepGLAD (Kong et al., 2024), which leverages latent diffusion models to generate full-atom peptide structures. For the scaffold generation task, we modify the inference script so that the input peptide includes certain known masked blocks instead of all blocks being masked. We maintain the same hyperparameter settings as described in the original paper.

# G ADDITIONAL RESULT

## G.1 TM-SCORE AND AAR

Table 4: TM-Score and Sequence Recovery Rate in peptide design and scaffold generation tasks.

| Table 5: Peptide Design Task | | | Table 6: Scaffold Generation Task | | |
|---|---|---|---|---|---|
| **Method** | **TM** | **AAR** | **Method** | **TM** | **AAR** |
| RFDiffusion | 0.44 | 40.14 | RFDiffusion (K=3) | 0.46 | 31.14 |
| ProteinGenerator | 0.43 | 45.82 | ProteinGenerator (K=3) | 0.48 | 32.05 |
| PepFlow | 0.38 | 51.25 | PepFlow (K=3) | 0.37 | 51.90 |
| PepGLAD | 0.29 | 20.59 | PepGLAD | 0.30 | 21.48 |
| PepHAR (K=1) | 0.33 | 32.32 | PepHAR (K=1) | 0.33 | 32.90 |
| PepHAR (K=2) | 0.32 | 39.91 | PepHAR (K=2) | 0.35 | 35.06 |
| PepHAR (K=3) | 0.34 | 34.36 | PepHAR (K=3) | 0.38 | 35.34 |

## G.2 CUMULATIVE ERROS

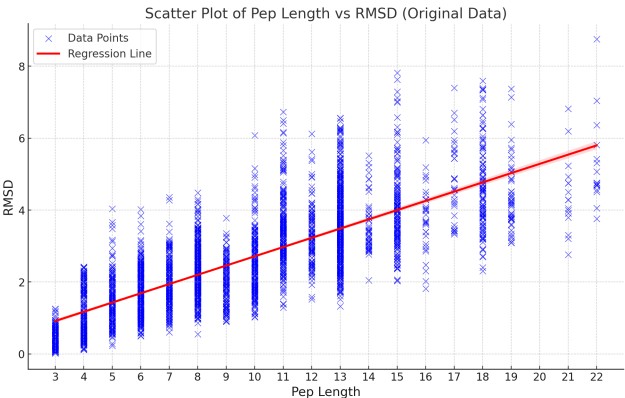

Figure 6: Scatter Plot of peptide length and RMSD value.

We analyzed the relationship between the length of the generated peptides and their corresponding RMSD values compared to native structures. As shown in the figure, peptide length demonstrates a strong positive correlation with RMSD, indicating that the autoregressive generation process accumulates errors as peptide length increases. Specifically, during the extension stage, dihedral angle predictions are used to iteratively add residues to the peptide structure. However, these predictions are not always precise. For example, when predicting the dihedral angles for a new residue, small deviations from the ground truth can occur. These deviations lead to slight inaccuracies in the reconstructed residue's backbone conformation. Once a residue is added with structural bias, the error propagates to subsequent extension steps. For instance, if the incorrect backbone conformation positions the residue slightly off from its ideal location, the next prediction step will use this biased structure as input. This can result in further deviation in the predicted dihedral angles for the next residue, introducing additional distortions to the growing peptide. Over multiple iterations, these small inaccuracies accumulate, leading to increasing structural deviations from the native conformation. This accumulation of structural errors explains the observed increase in RMSD with longer peptide sequences. The inherent dependency of the autoregressive process on previously generated fragments makes it particularly susceptible to error propagation during residue-by-residue extension. Addressing this limitation is crucial to improving the accuracy of peptide structure generation in future work.

Future work could address this accumulation issue in our autoregressive extension model. First, we can improve the precision of each prediction step by employing more accurate dihedral prediction models. Given the limited peptide datasets, leveraging pretrained models trained on larger datasets or incorporating traditional energy relaxation methods at each extension step to correct backbone bias could be beneficial. Second, advancements in autoregressive language modeling could be applied, such as predicting multiple dihedral angles simultaneously (Qi et al., 2020; Gloeckle et al., 2024), injecting noise during training (Pasini et al.), or using diffusion models to refine the predicted posterior distribution (Li et al., 2024c). Third, non-autoregressive approaches, such as generative modeling (Kenton & Toutanova, 2019; Chang et al., 2022) or diffusion models (Wu et al., 2024a), could be explored to generate all dihedral angles simultaneously and refine predictions iteratively.

## G.3 DIFFERENT K VALUES OF BASELINES

As shown in Table 7, unlike PepHAR, which benefits from increasing K in terms of geometry and energy metrics, the baseline models show only marginal improvements or even performance degradation. We believe this is due to their training schemes, which do not explicitly condition on known hotspots. Optimizing their training and inference processes for the scaffold setting could potentially improve their performance.

Table 7: Evaluation of methods in the scaffold generation task. $K = 1, 2, 3$ is the number of hot spots.

| | Valid % ↑ | RMSD Å ↓ | SSR % ↑ | BSR % ↑ | Stability % ↑ | Affinity % ↑ | Novelty % ↑ | Diversity % ↑ | Success % ↑ |
|---|---|---|---|---|---|---|---|---|---|
| RFDiffusion ($K = 1$) | 66.80 | 3.51 | 63.19 | 23.56 | 17.73 | 20.58 | 66.17 | 22.46 | 19.19 |
| RFDiffusion ($K = 2$) | 68.68 | 2.85 | 65.68 | 31.14 | 18.69 | 22.34 | 45.53 | 24.14 | 21.59 |
| RFDiffusion ($K = 3$) | **69.88** | 4.09 | 63.66 | 26.83 | 20.07 | 21.26 | 55.03 | 26.67 | 23.15 |
| ProteinGenerator ($K = 1$) | 68.00 | 3.79 | 64.53 | 25.52 | 18.59 | 20.55 | 60.39 | 21.28 | 20.04 |
| ProteinGenerator ($K = 2$) | 69.20 | 3.92 | 66.79 | 30.61 | 19.08 | 21.54 | 55.24 | 23.29 | 20.42 |
| ProteinGenerator ($K = 3$) | 68.52 | 3.95 | 65.86 | 24.17 | 20.40 | **22.80** | 50.73 | 20.82 | 24.90 |
| PepFlow ($K = 1$) | 40.35 | 2.51 | 79.58 | 86.40 | 10.55 | 18.13 | 50.46 | 16.29 | **26.46** |
| PepFlow ($K = 2$) | 49.29 | 2.82 | 79.23 | 85.05 | 10.19 | 18.20 | 54.74 | 19.52 | 24.03 |
| PepFlow ($K = 3$) | 42.68 | 2.45 | 81.00 | 82.76 | 11.17 | 13.64 | 50.93 | 16.97 | 24.54 |
| PepGLAD ($K = 1$) | 53.45 | 3.87 | 76.59 | 20.15 | 14.54 | 12.03 | 50.46 | 30.84 | 13.56 |
| PepGLAD ($K = 2$) | 52.96 | 3.93 | 75.06 | 20.02 | 10.29 | 15.72 | 54.74 | 30.36 | 14.03 |
| PepGLAD ($K = 3$) | 53.51 | 3.84 | 76.26 | 19.61 | 12.22 | 18.27 | 50.93 | **30.99** | 14.85 |
| PepHAR ($K = 1$) | 56.01 | 3.72 | 80.61 | 78.18 | 17.89 | 19.94 | **80.61** | 29.79 | 20.43 |
| PepHAR ($K = 2$) | 55.36 | 2.85 | 82.79 | 85.80 | 19.18 | 19.17 | 74.76 | 25.32 | 22.09 |
| PepHAR ($K = 3$) | 55.41 | **2.15** | **83.02** | **88.02** | **20.50** | 20.65 | 72.56 | 19.68 | 21.45 |

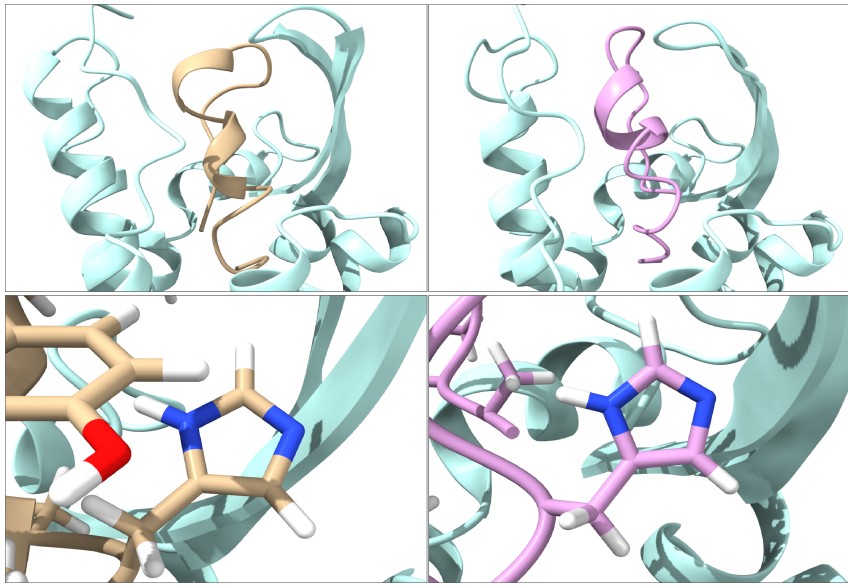

Figure 7: Upper Left: Human Endothelin type B receptor in complex with the ET1 peptide binder. Upper Right: Receptor in complex with PepHAR-generated peptide binder. Lower Left: HIS16 in the ET1 peptide is identified as a hotspot residue. Lower Right: PepHAR recovers this hotspot residue in the peptide design task.

### G.4 CASE STUDY ON GPCR-PEPTIDE INTERACTION

Here, we present a case study of generating a peptide binder for the human Endothelin type B receptor (PDB: 5GLH) (Shihoya et al., 2018). As shown in Fig. 7, the designed peptide exhibits secondary structures (helix) similar to the native peptide binder (ET1 peptide), interacting with the receptor residues at the top and within the receptor. Through per-residue energy calculations and manual inspection, the HIS16 residue in the ET1 peptide is identified as a hotspot residue, playing a crucial role in contacting the receptor's extracellular loop regions. Remarkably, PepHAR recovers this HIS16 residue in the generated peptide. The orientation of the functional group in the generated HIS16 closely resembles that of the native hotspot residue, demonstrating that PepHAR can effectively recover hotspot interactions during the design process.

### G.5 SECONDARY STRUCTURE ANALYSIS

Table 8: Secondary structure composition evaluation.

| Method | Coil % | Helix % | Strand % |
|---|---|---|---|
| Native | 75.20 | 11.47 | 13.15 |
| PepHAR ($K = 3$) | 89.42 | 10.36 | 0.21 |
| Hotspots | 90.13 | 9.48 | 0.22 |

We analyzed the secondary structure proportions of native peptides, generated peptides, and hotspot residues in the test set, as shown in Table 8.

Compared to native peptides, PepHAR-generated peptides and hotspots tend to exhibit a higher proportion of coil regions (bonded turns, bends, or loops) while maintaining similar proportions of helix regions. However, the proportion of strand regions is significantly lower compared to native peptides, indicating that strand regions in the native structure are often replaced by coil regions in the generated peptides. This could be due to subtle differences in structural parameters required for forming strands. We believe that further energy relaxation using tools like Rosetta, OpenMM, or FoldX could help refine the peptide structures to form more accurate secondary structures.

Regarding hotspot occurrences, we observe that hotspots are predominantly located in coil and helix regions, with almost no presence in strand regions. This aligns with interaction principles, where strand regions may represent structural transitions, while the irregular coil regions or the relatively stable helix regions are more likely to serve as functional interaction sites.

## H CURRENT WORKS ON PEPTIDE DESIGN METHOD

Recent advancements in computational peptide binder design have significantly benefited from deep learning. Bryant & Elofsson (2022) introduced EvoBind, an in silico directed evolution platform that utilizes AlphaFold to design peptide binders targeting specific protein interfaces using only sequence information. Building on this, Li et al. (2024b) developed EvoBind2, which extends EvoBind's capabilities by enabling the design of both linear and cyclic peptide binders of varying lengths solely from a protein target sequence, without requiring the specification of binding sites or binder sizes. Using MCTS simulations and reinforcement learning,Wang et al. (2023) engineers peptide binders and achieves comparable result to EvoBind. Chen et al. (2023) proposed PepMLM, a target-sequence-conditioned generator for linear peptide binders based on masked language modeling. By employing a novel masking strategy, PepMLM effectively reconstructs binder regions, achieving low perplexities and demonstrating efficacy in cellular models. Leveraging geometric convolutional neural networks to decipher interaction fingerprints from protein interaction surfaces, Gainza et al. (2020) developed MaSIF. Subsequently, BindCraft (Pacesa et al., 2024) combined AF2-Multimer sampling with ProteinMPNN sequence design to optimize protein-protein interaction surfaces. Similar to our approach of defining key hotspot residues, AlphaProteo (Zambaldi et al., 2024) also aims to generate high-affinity protein binders that specifically interact with designated residues on the target protein. PepGLAD (Kong et al., 2024) encodes full-atom peptide structures and sequences using an equivariant graph neural network, and applies latent diffusion on the peptide embedding space to explore peptide generation.

