# OpenReview forum: "Hotspot-Driven Peptide Design via Multi-Fragment Autoregressive Extension"
_ICLR.cc/2025/Conference — ICLR 2025 Poster_

### Official Review · Reviewer_Gac7 · 2024-10-27

**Soundness:** 4
**Presentation:** 3
**Contribution:** 4
**Rating:** 8
**Confidence:** 3

**Summary:**

The authors present a novel de-novo peptide design method, PepHAR, aimed at generating binders for specific sites on a target protein. The method leverages the concept of "hotspots", a small number of residues at protein-protein interfaces that contribute most of the binding energy, with other residues acting as "scaffolds" to stabilize the interface. PepHAR’s peptide prediction process is based on three main components: (1) A neural network trained using a noise contrastive estimation (NCE) loss to differentiate between real interface residues (ground truth) and noisy counterparts altered with spatial perturbations. The authors use Langevin dynamics to sample from this energy-based model. (2) A second neural network that builds a polypeptide chain autoregressively by predicting the phi-psi dihedral angles for each subsequent residue. This model, trained using maximum likelihood estimation (MLE), predicts angles for residues extending either direction along the chain. (3) A final correction step to refine the predicted peptide structures, addressing minor inaccuracies like dihedral angle errors, steric clashes, or chain breaks. This stage optimizes dihedrals based on neighboring residues and rebuilds the peptide, comparing it to the initial prediction. An objective function minimizes the distance between the rebuilt and predicted structures while ensuring angles align with the learned dihedral distribution.
PepHAR was evaluated for de-novo peptide generation with up to three hotspots, comparing favorably to other leading methods like RFdiffusion, PepFlow, and ProteinGenerator. An ablation study also confirmed the significance of each method component. Overall, PepHAR generated peptides with lower RMSD to the native structure than other methods, showing its effectiveness in autoregressively designing peptides for target protein interfaces.

**Strengths:**

The paper is well-written, with ideas and concepts clearly explained. Given the ongoing challenges in de-novo peptide binder design, this novel and original approach is both relevant and significant. I especially appreciated the clear presentation of the derivations and fundamental principles underlying the neural network design and training.

**Weaknesses:**

The paper is well-written, with few notable weaknesses.

**Questions:**

I have two minor questions, primarily out of curiosity:

(1) In the benchmarks where PepHAR was tested, could you provide more insights into the nature of the predicted structures? For instance, do the predictions tend to cluster around helical bundles, or is there a broader range of conformations? Additionally, are the k=3 hotspots located within disordered regions, or are they typically in helices, where the backbone might be easier to infill? I would suggest to include a brief analysis of the secondary structure composition of the generated peptides and compare it to native structures (or a representative set of native structures), as well as an examination of where the hotspots tend to be located in terms of secondary structure elements.

(2) When multiple hotspots are specified, fragments are grown from each hotspot. How does the network ensure that these initially disconnected fragments will eventually fit together? Is this addressed during the extension phase or in the polishing stage? Furthermore, is there a risk that extended chains from different fragments might fail to converge into a closed structure? It could be useful to elaborate on the fragment assembly method, and add statistics on how often fragments successfully connect and if there are any strategies used to ensure proper assembly or safeguard this process.

---

> ### Author Response · Authors · 2024-11-19
> **Response to Reviewer Gac7**
>
> Thank you for your insightful and constructive comments, as well as your appreciation of our work in ** novel approach, clear presentation and fundamental principles underlying network design and training **. Below, we provide clarifications and responses to your questions. If any concerns remain unaddressed, please feel free to reach out with further inquiries; we would be happy to engage in additional discussions.
>
>
> ## Q1: I would suggest to include a brief analysis of the secondary structure composition of the generated peptides and compare it to native structures (or a representative set of native structures), as well as an examination of where the hotspots tend to be located in terms of secondary structure elements.
>
> We analyzed the secondary structure proportions of native peptides, generated peptides, and hotspot residues in the test set. The results are as follows:
>
> | Method        	| Coil % | Helix % | Strand % |
> |--------------------|--------|---------|----------|
> | Native        	| 75.20  | 11.47   | 13.15	|
> | PepHAR ($K=3$)	| 89.42  | 10.36   | 0.21 	|
> | Hotspots      	| 90.13  | 9.48	| 0.22 	|
>
> Compared to native peptides, PepHAR-generated peptides and hotspots tend to exhibit a higher proportion of coil regions (bonded turns, bends, or loops) while maintaining similar proportions of helix regions. However, the proportion of strand regions is significantly lower compared to native peptides, indicating that strand regions in the native structure are often replaced by coil regions in the generated peptides. This could be due to subtle differences in structural parameters required for forming strands. We believe that further energy relaxation using tools like Rosetta, OpenMM, or FoldX could help refine the peptide structures to form more accurate secondary structures.
>
> Regarding hotspot occurrences, we observe that hotspots are predominantly located in coil and helix regions, with almost no presence in strand regions. This aligns with interaction principles, where strand regions may represent structural transitions, while the irregular coil regions or the relatively stable helix regions are more likely to serve as functional interaction sites.
>
> ## Q2: When multiple hotspots are specified, fragments are grown from each hotspot. How does the network ensure that these initially disconnected fragments will eventually fit together? Is this addressed during the extension phase or in the polishing stage? It could be useful to elaborate on the fragment assembly method, and add statistics on how often fragments successfully connect and if there are any strategies used to ensure proper assembly or safeguard this process.
>
>
> The network addresses the assembly of disconnected fragments through mechanisms implemented in both the extension and correction (polishing) stages:
>
> + Extension Stage: During this phase, each fragment's residue generation is conditioned not only on its own sequence and structure but also on the fragments generated earlier. The network uses structural and sequential information from previously generated fragments to guide the placement of new residues, helping align the fragments in 3D space. This is achieved through training on a combined structure and sequence reconstruction loss, which enforces the network to learn how to grow fragments in a way that facilitates eventual connectivity.
>
> + Correction Stage: In this phase, residues at the junctions between fragments, as well as other intermediate residues, are refined to ensure seamless assembly. The correction stage enforces constraints on positional, orientational, and dihedral angle consistency, enabling previously disconnected fragments to align and fit together properly.
>
> The success rate of fragment assembly can be indirectly measured using the valid metric, which evaluates whether the inter-residue distances fall within acceptable ranges. As demonstrated in Table 3, the correction stage significantly improves these metrics by refining the geometry and resolving inconsistencies introduced during the extension phase. This suggests that the network effectively resolves fragment connectivity issues through a combination of structural alignment and dihedral angle optimization.

---

> > ### Comment · Reviewer_Gac7 · 2024-11-25
> >
> > Thank you for addressing my concerns. It is now much clearer how the fragments are connected when conditioned on each other. This is an excellent paper, and I will maintain my high score for it.

---

> > > ### Author Response · Authors · 2024-11-26
> > >
> > > Thank you for your thoughtful and generous feedback—it means a great deal to us. We are truly grateful for your recognition and are honored by your high evaluation of our work!

---

### Official Review · Reviewer_4QbP · 2024-10-28

**Soundness:** 2
**Presentation:** 3
**Contribution:** 3
**Rating:** 6
**Confidence:** 5

**Summary:**

The paper introduces PepHAR, a state-of-the-art generative model for peptide design. It employs an energy-based density model to precisely sample key residues, extends peptides with an autoregressive approach based on dihedral angle estimation, and refines fragment assembly through an iterative optimization process. The paper proposes a new experimental setting, scaffold generation, to mimic practical scenarios, and demonstrate the competitive performance of PepHAR in both peptide design and scaffold generation tasks.

**Strengths:**

- The paper innovatively integrates hotspot residues into peptide design, a concept well-supported by biological theories, enhancing the model's relevance and novelty in the field.
- The three-stage approach, including the use of dihedral angles and the correction stage, is a robust technical framework that ensures the generation of structurally sound peptides.
- The model demonstrates excellent performance in SSR, Affinity, Novelty, and Diversity, indicating its ability to produce high-quality, functional, and diverse peptides.

**Weaknesses:**

1. **Performance in Specific Metrics**: As shown in Section 5.1, Table 1, PepHAR underperforms RFDiffusion in the Valid metric by a significant margin (10%). This suggests that PepHAR may not always generate peptides with geometries that are fully consistent with the formation of peptide bonds. Additionally, PepHAR's Affinity metric indicates room for improvement, potentially due to the density model's limitations in sufficiently approximating the complex energy landscapes involved in peptide-target interactions.
2. **Limited Exploration in Scaffold Generation**: The scaffold generation scenario only considers a small number of hotspots (K=1, 2, 3), with baseline comparisons limited to K=3. This narrow exploration may not capture the full spectrum of possible hotspot configurations, potentially limiting the generalizability of the findings to more complex scenarios.
3. **Density Model Limitations**: The energy-based density model used in PepHAR might not fully approximate the intricate energy landscapes of peptide interactions, which could impact the model's ability to generate peptides with optimal binding affinities.

**Questions:**

1. In Section 5.2, Table 2, it would be insightful to see the performance of the baseline models under different conditions, specifically for K=1 and K=2, which are currently not included. Additionally, to thoroughly understand the influence of hotspot count on peptide design and the tradeoff with diversity mentioned in Section 5.3, it would be valuable to conduct experiments with a broader range of K values, or explain the insights behind the choice of these values.
2. The current study employs RFDiffusion and PepFlow as baseline models. Given the recent advancements in diffusion or flow-based peptide design, it would be valuable to include additional baseline models such as PepGLAD
3. Why does PepHAR underperform RFDiffusion and PepFlow in metrics such as valid% and RMSD, and what is the tradeoff involved?
4. The paper mentions tasks such as HIV-1+CD4 and p53+MDM2 interactions, which are known for their hotspot-mediated binding. It would be beneficial to see PepHAR's performance demonstrated on these specific systems.

---

> ### Author Response · Authors · 2024-11-19
> **Response to Reviewer 4QbP, Part 1**
>
> Thank you for your insightful and constructive comments, as well as your appreciation of our work in **innovatively integrating hotspot residues into peptide design and proposing a robust technical framework**. Below, we provide clarifications and responses to your questions. If any concerns remain unaddressed, please feel free to reach out with further inquiries; we would be happy to engage in additional discussions.
>
> ## Q1: The scaffold generation scenario only considers a small number of hotspots (K=1, 2, 3), with baseline comparisons limited to K=3. it would be insightful to see the performance of the baseline models under different conditions, specifically for K=1 and K=2, which are currently not included. it would be valuable to conduct experiments with a broader range of K values, or explain the insights behind the choice of these values.
>
> We chose K=1, 2, 3 based on the peptide length distribution in our training and test datasets, where 3 represents the minimum peptide length.
>
> To address the impact of varying K values on baseline models in the scaffold generation task, we conducted additional experiments, as detailed in Appendix G.3. The updated results are summarized below. As shown, unlike PepHAR, which benefits from increasing K in terms of geometry and energy metrics, the baseline models show only marginal improvements or even performance degradation. We believe this is due to their training schemes, which do not explicitly condition on known hotspots. Optimizing their training and inference processes for the scaffold setting could potentially improve their performance.
>
> ***Peptide Scaffold Generation***
>
> | Method                     | Valid % ↑ | RMSD Å ↓ | SSR % ↑ | BSR % ↑ | Stability % ↑ | Affinity % ↑ | Novelty % ↑ | Diversity % ↑ | Success % ↑ |
> |----------------------------|-----------|----------|---------|---------|---------------|--------------|-------------|---------------|-------------|
> | RFDiffusion (K=1)          | 66.80     | 3.51     | 63.19   | 23.56   | 17.73         | 20.58        | 66.17       | 22.46         | 19.19       |
> | RFDiffusion (K=2)          | 68.68     | 2.85     | 65.68   | 31.14   | 18.69         | 22.34        | 45.53       | 24.14         | 21.59       |
> | RFDiffusion (K=3)          | **69.88** | 4.09     | 63.66   | 26.83   | 20.07         | 21.26        | 55.03       | 26.67         | 23.15       |
> | ProteinGenerator (K=1)     | 68.00     | 3.79     | 64.53   | 25.52   | 18.59         | 20.55        | 60.39       | 21.28         | 20.04       |
> | ProteinGenerator (K=2)     | 69.20     | 3.92     | 66.79   | 30.61   | 19.08         | 21.54        | 55.24       | 23.29         | 20.42       |
> | ProteinGenerator (K=3)     | 68.52     | 3.95     | 65.86   | 24.17   | 20.40         | **22.80**    | 50.73       | 20.82         | 24.90       |
> | PepFlow (K=1)              | 40.35     | 2.51     | 79.58   | 86.40   | 10.55         | 18.13        | 50.46       | 16.29         | **26.46**   |
> | PepFlow (K=2)              | 49.29     | 2.82     | 79.23   | 85.05   | 10.19         | 18.20        | 54.74       | 19.52         | 24.03       |
> | PepFlow (K=3)              | 42.68     | 2.45     | 81.00   | 82.76   | 11.17         | 13.64        | 50.93       | 16.97         | 24.54       |
> | PepGLAD (K=1)              | 53.45     | 3.87     | 76.59   | 20.15   | 14.54         | 12.03        | 50.46       | 30.84         | 13.56       |
> | PepGLAD (K=2)              | 52.96     | 3.93     | 75.06   | 20.02   | 10.29         | 15.72        | 54.74       | 30.36         | 14.03       |
> | PepGLAD (K=3)              | 53.51     | 3.84     | 76.26   | 19.61   | 12.22         | 18.27        | 50.93       | **30.99**     | 14.85       |
> | PepHAR (K=1)               | 56.01     | 3.72     | 80.61   | 78.18   | 17.89         | 19.94        | **80.61**   | 29.79         | 20.43       |
> | PepHAR (K=2)               | 55.36     | 2.85     | 82.79   | 85.80   | 19.18         | 19.17        | 74.76       | 25.32         | 22.09       |
> | PepHAR (K=3)               | 55.41     | **2.15** | **83.02**| **88.02**| **20.50**    | 20.65        | 72.56       | 19.68         | 21.45       |

---

> > ### Author Response · Authors · 2024-11-19
> > **Response to Reviewer 4QbP, Part 2**
> >
> > ## Q2: Given the recent advancements in diffusion or flow-based peptide design, it would be valuable to include additional baseline models such as PepGLAD.
> >
> > We appreciate this suggestion. PepGLAD [1] leverages latent diffusion models for full-atom peptide structure and sequence generation. Using the official implementation and checkpoint, we modified the inference script to support scaffold generation. We have added PepGLAD results to Tables 1 and 2 in the main text, as summarized below:
> >
> > ***Peptide Binder Design***
> >
> > | Method          	| Valid % ↑ | RMSD Å ↓  | SSR % ↑  | BSR % ↑  | Stability % ↑ | Affinity % ↑ | Novelty % ↑  | Diversity % ↑ |
> > |---------------------|-----------|-----------|----------|----------|----------------|--------------|--------------|---------------|
> > | PepGLAD         	| 55.20 	| 3.83  	| 80.24	| 19.34	| 20.39      	| 10.47    	| 75.07    	| 32.10     	|
> > | PepHAR ($K=1$)  	| 57.99 	| 3.73  	| 79.93	| 84.17	| 15.69      	| 18.56    	| 81.21	| 32.69 	|
> > | PepHAR ($K=2$)  	| 55.67 	| 3.19  	| 80.12	| 84.57	| 15.91      	| 19.82    	| 79.07    	| 31.57     	|
> > | PepHAR ($K=3$)  	| 59.31 	| 2.68  	| **84.91**| 86.74	| 16.62      	| 20.53    	| 79.11    	| 29.58     	|
> >
> > ***Peptide Scaffold Generation***
> >
> > | Method           	| Valid % ↑ | RMSD Å ↓  | SSR % ↑  | BSR % ↑  | Stability % ↑ | Affinity % ↑ | Novelty % ↑  | Diversity % ↑  | Success % ↑ |
> > |-----------------------|-----------|-----------|----------|----------|----------------|--------------|--------------|----------------|-------------|
> > | PepGLAD ($K=3$)  	| 53.51 	| 3.84  	| 76.26	| 19.61	| 12.22      	| 18.27    	| 50.93    	| 30.99  	| 14.85   	|
> > | PepHAR ($K=1$)   	| 56.01 	| 3.72  	| 80.61	| 78.18	| 17.89      	| 19.94    	| 80.61	| 29.79  	| 20.43   	|
> > | PepHAR ($K=2$)   	| 55.36 	| 2.85  	| 82.79	| 85.80	| 19.18      	| 19.17    	| 74.76    	| 25.32      	| 22.09   	|
> > | PepHAR ($K=3$)   	| 55.41 	| 2.15  | 83.02| 88.02| 20.50  	| 20.65    	| 72.56    	| 19.68      	| 21.45   	|
> >
> > ## Q3: Why does PepHAR underperform RFDiffusion and PepFlow in metrics such as valid% and RMSD, and what is the tradeoff involved?
> >
> > As shown in our experiments, RFDiffusion and ProteinGenerator generate peptides with more valid geometries and stable energies, consistent with the findings in PepFlow [2]. We speculate this is due to their use of RosettaFold pretrained weights and training on much larger PDB datasets, which enables them to generalize well and effectively capture native protein distributions.
> >
> > For the comparison between PepFlow and PepHAR, the discrepancy might arise because flow-matching methods are generally better at fitting data distributions compared to EBM models trained with NCE loss, as they are easier to train and sample from. Further improvement to PepHAR could be achieved by carefully optimizing its training and inference processes. Additionally, as you pointed out, the energy-based density model might not fully capture intricate energy landscapes. We propose two future directions to address this limitation: (1) exploring more efficient parameterizations of EBM distributions—PepHAR currently uses $\frac{1}{Z} \exp\left(g_{\theta, c}(x, O \mid T)\right)$ as the energy function, and alternative forms could improve consistency and relationships between structure and sequence [3,4]; (2) adopting advanced training techniques to enhance performance [5,6].
> >
> > It is also worth noting that RMSD is a double-edged metric. If the goal is to design peptides similar to native ones, lower RMSD is preferable. However, for designing diverse peptides targeting the same receptors, a higher RMSD may indicate more innovation. To balance this, RMSD should remain within an acceptable range to ensure the generated peptides are not too far from the native conformation, preserving their binding ability and functionality.
> >
> > ## Q4: The paper mentions tasks such as HIV-1+CD4 and p53+MDM2 interactions, which are known for their hotspot-mediated binding. It would be beneficial to see PepHAR's performance demonstrated on these specific systems.
> >
> > In the appendix, we provide examples to illustrate the definition of hotspot residues. For case studies, some generated peptide examples are presented in Figures 4 and 5 of the draft. Additionally, we include another specific case study on GPCR-peptide interactions in Appendix G.4 (Figure 7). This example demonstrates that PepHAR can generate peptides with similar secondary structures, binding sites, and hotspot residues compared to native peptides. As shown in Figure 7, PepHAR successfully samples hotspot residues that interact effectively with receptors.
> >
> > We believe PepHAR has the potential to be applied to the design of peptides for other biologically important systems, and we look forward to seeing future work utilizing our tool for wet-lab experiments.

---

> > > ### Author Response · Authors · 2024-11-19
> > > **Response to Reviewer 4QbP, Part 3**
> > >
> > > ## References:
> > >
> > > [1] Kong, Xiangzhe, et al. "Full-atom peptide design with geometric latent diffusion." arXiv preprint arXiv:2402.13555 (2024).
> > >
> > > [2] Li, Jiahan, et al. "Full-Atom Peptide Design based on Multi-modal Flow Matching." arXiv preprint arXiv:2406.00735 (2024).
> > >
> > > [3] Ren, Milong, et al. "Accurate and robust protein sequence design with CarbonDesign." Nature Machine Intelligence 6.5 (2024): 536-547.
> > >
> > > [4] Ren, Milong, Tian Zhu, and Haicang Zhang. "CarbonNovo: Joint Design of  Protein Structure and Sequence Using a Unified Energy-based Model." Forty-first International Conference on Machine Learning.
> > >
> > > [5] Du Y, Mordatch I. Implicit generation  and modeling with energy based models[J]. Advances in Neural Information Processing Systems, 2019, 32.
> > >
> > > [6] Song, Yang, and Diederik P. Kingma. "How to train your energy-based models." arXiv preprint arXiv:2101.03288 (2021).

---

> ### Comment · Reviewer_4QbP · 2024-11-23
> **my concerns have been adequately addressed by the authors**
>
> Thank you for your comprehensive reply to my previous review.You have provided a clear explanation regarding the points I raised. I will maintain my score of 6.  I look foward to seeing your work  with wider training dataset and incorporating wet lab validations beyond traditional metrics.

---

> > ### Author Response · Authors · 2024-11-23
> > **Response to Reviewer 4QbP**
> >
> > We sincerely appreciate the time and effort you have devoted to assisting us in refining our work, particularly your suggestion to study how varying the number of hot spots affects baseline results. This has been incredibly insightful for us. In our future work, we plan to curate a larger binder-receptor dataset by incorporating protein fragment interactions for data augmentation, along with more advanced in silico results. Additionally, we aim to gather published experimental peptide results to retrospectively evaluate different methods and apply our approaches to design peptides for wet lab testing (especially GPCR peptides).
> >
> > If you find our responses satisfactory, we kindly request that you consider revising your initial assessment. Please feel free to share any further insights or suggestions you may have—we greatly value your feedback.

---

### Official Review · Reviewer_mmeW · 2024-10-30

**Soundness:** 2
**Presentation:** 3
**Contribution:** 2
**Rating:** 5
**Confidence:** 4

**Summary:**

The paper introduces PepHAR, a hot-spot-driven autoregressive generative model for designing peptides targeted at specific protein binding sites. The authors propose a three-stage approach: 1) hotspot residue sampling using an energy-based model, 2) fragmented autoregressive extension of peptide structure using dihedral angles, and 3) correction through gradient-based optimization to ensure overall peptide geometry validity. Through experiments, PepHAR demonstrates competitive performance in scaffold-based peptide design, showing improvements in structure and binding accuracy when compared to state-of-the-art models like RFDiffusion and PepFlow on metrics including binding site overlap, novelty, and stability.

**Strengths:**

The paper attempt to solve an important protein design problem with significant biomedical applications.

The hotspot-driven approach combined with an autoregressive fragment assembly model is novel within peptide design, providing a unique perspective on addressing protein-peptide design problem.

The paper offers a detailed breakdown of the approach, supported by schematics and visualizations that help clarify the generation process.

The paper uses comprehensive metrics (e.g., validity, stability, affinity, diversity, novelty), which cover structural, energetic, and novelty aspects of peptide design.

**Weaknesses:**

- The autoregressive fragment extension process might introduces cumulative geometric errors, especially for longer peptides.

-  Although PepHAR outperforms some baselines, the reported improvements are incremental and may not justify the added model complexity and limitations. For a conditional generation task, the primary metric is RMSD, which quantifies the structural similarity between the generated structures and native structures. where PepHAR is worse than the baseline PepFlow.

- The claim of the paper is purely using the frame representation for peptide might break the validity of the protein structures like bond formulation. However, Table 1 shows that RFDiffusion, which is a frame-representation-based method, exhibits higher Valid %.

- I would expect open source code to support the methodology and experimental results.

- Regarding the equation, the author should add period to the end if it closes a sentence. I notice that the author missed that in most of the equations in this paper.

**Questions:**

- I expect the author to quantify the impact of the cumulative errors from autoregressive generation.
- Figure 4 shows the designed peptides are all loops. Given that, I wonder if the SSR % is still meaningful if most of the native and designed peptides are just loops. Because in that case, the model can just produce all loopy structures and get a very high SSR.
- I suggest that the author further design the sequences for those generated peptide structures using for example, ProteinMPNN and report the recovery rate. This could reflect the peptide structure quality since low-quality structures would result in low sequence recovery or low-quality sequences generated such as repetitions.

---

> ### Author Response · Authors · 2024-11-19
> **Response to Reviewer mmeW**
>
> Thank you for your insightful and constructive comments as well as your appreciation of our work that **combines energy-based hotspot-driven model with autoregressive fragment assembly and using detailed breakdown and comprehensive metrics**. Below are some clarifications and answers to your questions. If our response does not fully address your concerns, please post additional questions; we will be happy to discuss further.
>
> ## Q1: The autoregressive fragment extension process might introduces cumulative geometric errors, especially for longer peptides. I expect the author to quantify the impact of the cumulative errors from autoregressive generation.
>
> We have quantified the cumulative geometric errors in Appendix Fig. 6, subsection G.2. As shown in the figure, peptide length exhibits a strong positive correlation with RMSD, indicating that generating longer peptides results in more cumulative errors due to the autoregressive generation process.
>
> ## Q2: Regarding the equation, the author should add period to the end if it closes a sentence. I notice that the author missed that in most of the equations in this paper.
>
> We have added commas and periods to the equations where appropriate, with commas for equations within sentences and periods for those that conclude sentences.
>
> ## Q3: Figure 4 shows the designed peptides are all loops. Given that, I wonder if the SSR % is still meaningful if most of the native and designed peptides are just loops. Because in that case, the model can just produce all loopy structures and get a very high SSR.
>
> The peptide examples in Figures 4 and 5 represent loop regions, as cyclic peptides are among the most promising peptide mediators [1,2,3]. However, helical peptides, such as GPCR peptides, are also representative peptide structures [4,5], and evaluating the secondary structures of peptides remains a widely studied topic [6,7], as shown in previous works [8]. To provide additional context, we calculated the secondary structure labels for the native peptides in the test set, as follows:
>
> | Secondary Structure        	| Proportion (%) |
> |--------------------------------|----------------|
> | Loops and irregular elements   | 58.20      	|
> | Bend                       	| 7.58       	|
> | Extended strand            	| 11.42      	|
> | Hydrogen-bonded turn       	| 9.42       	|
> | 3-helix (3/10 helix)       	| 0.70       	|
> | Alpha helix                	| 10.77      	|
> | Residue in isolated beta-bridge| 1.73       	|
> | Others                     	| 0.16       	|
>
> Although 58% of residues are loops or irregular regions, other secondary structures, particularly beta strands and alpha helices, are also present. Both loop regions and helical/strand regions play crucial roles in protein interactions, making it important to generate residue structures at precise positions with accurate secondary structures. We believe that assessing the secondary structure rate (SSR %) remains a valuable metric for evaluating the quality of peptide design, which is also conducted in the prior work. We also include another example in Appendix Figure 7 to show that our generated peptides can also have helix structures.
>
> ## Q4: I suggest that the author further design the sequences for those generated peptide structures using for example, ProteinMPNN and report the recovery rate.
>
> We have added the sequences and recovery rates for each example in Figures 4 and 5. In general, the lower recovery rates indicate structural differences compared to the native structures.
>
> ## References:
> [1] Design of linear and  cyclic peptide binders of different lengths only from a protein target  sequence. bioRxiv, 2024-06
>
> [2] Hosseinzadeh, Parisa, et al. "Comprehensive computational design of ordered peptide macrocycles." Science 358.6369 (2017): 1461-1466.
>
> [3] Bhardwaj, Gaurav, et al. "Accurate de novo design of membrane-traversing macrocycles." Cell 185.19 (2022): 3520-3532.
>
> [4] Pal, Kuntal, Karsten Melcher, and H. Eric Xu. "Structure and mechanism  for recognition of peptide hormones by Class B G-protein-coupled  receptors." Acta pharmacologica sinica 33.3 (2012): 300-311.
>
> [5] London, Nir, Dana Movshovitz-Attias, and Ora Schueler-Furman. "The structural basis of peptide-protein binding strategies." Structure 18.2 (2010): 188-199.
>
> [6] Eiríksdóttir, Emelía, et al. "Secondary structure of cell-penetrating peptides controls membrane interaction and insertion." Biochimica et Biophysica Acta (BBA)-Biomembranes 1798.6 (2010): 1119-1128.
>
> [7] Seebach, Dieter, David F. Hook, and Alice Glättli. "Helices and other secondary structures of β‐and γ‐peptides." Peptide Science: Original Research on Biomolecules 84.1 (2006): 23-37.
>
> [8] Li, Jiahan, et al. "Full-Atom Peptide Design based on Multi-modal Flow Matching." arXiv preprint arXiv:2406.00735 (2024).

---

> > ### Comment · Reviewer_mmeW · 2024-11-21
> > **the authors have addressed my concerns**
> >
> > Thank you for the nice response! My concerns have been addressed. I appreciate that the authors can openly acknowledge the limitations in response 1. Would you like to add a bit more discussion about the accumulation of errors and future work on how to overcome/optimize them?

---

> > > ### Author Response · Authors · 2024-11-22
> > > **Follow Up Response: Addressing Accumulation Errors: Insights and Future Directions**
> > >
> > > Thank you for your follow-up questions regarding accumulation errors and potential future improvements. We have added more detailed discussions and future directions in Appendix Section G.2.
> > >
> > > As shown in the figure, peptide length demonstrates a strong positive correlation with RMSD, indicating that the autoregressive generation process accumulates errors as peptide length increases. Specifically, during the extension stage, ***dihedral angle predictions are used to iteratively add residues to the peptide structure. However, these predictions are not always precise***. For example, when predicting the dihedral angles for a new residue, small deviations from the ground truth can occur. These deviations lead to slight inaccuracies in the reconstructed residue’s backbone conformation.
> > >
> > > Once a residue is added with structural bias, the error propagates to subsequent extension steps. For instance, if the incorrect backbone conformation positions the residue slightly off from its ideal location, the next prediction step will use this biased structure as input. This can result in further deviation in the predicted dihedral angles for the next residue, introducing additional distortions to the growing peptide. ***Over multiple iterations, these small inaccuracies accumulate, leading to increasing structural deviations from the native conformation***.
> > >
> > > This accumulation of structural errors explains the observed increase in RMSD with longer peptide sequences. The inherent dependency of the autoregressive process on previously generated fragments makes it particularly susceptible to error propagation during residue-by-residue extension. Addressing this limitation is crucial to improving the accuracy of peptide structure generation in future work.
> > >
> > > To address these accumulation errors, several future improvements could be explored:
> > > 1. ***Improving Prediction Accuracy***: Enhancing the precision of each prediction step by using more accurate dihedral prediction models. Given the limited size of peptide datasets, leveraging pretrained models trained on larger datasets or incorporating traditional energy relaxation methods at each extension step to correct backbone bias could be effective.
> > > 2. ***Advancing Autoregressive Techniques***: Applying advanced techniques from autoregressive language modeling, such as predicting multiple dihedral angles simultaneously [1,2], injecting noise during training [3], or utilizing diffusion models to refine posterior distributions [4].
> > > 3. ***Exploring Non-Autoregressive Approaches***: Employing non-autoregressive methods, such as generative modeling [5,6] or diffusion models [7], to generate all dihedral angles simultaneously and iteratively refine predictions.
> > >
> > > ### References:
> > >
> > > [1] Qi, Weizhen, et al. "Prophetnet: Predicting future n-gram for sequence-to-sequence pre-training." *arXiv preprint arXiv:2001.04063* (2020).
> > >
> > > [2] Gloeckle, Fabian, et al. "Better & faster large language models via multi-token prediction." *arXiv preprint arXiv:2404.19737* (2024).
> > >
> > > [3] Pasini, Marco, et al. "Continuous Autoregressive Models with Noise Augmentation Avoid Error Accumulation." *Audio Imagination: NeurIPS 2024 Workshop AI-Driven Speech, Music, and Sound Generation*.
> > >
> > > [4] Li, Tianhong, et al. "Autoregressive Image Generation without Vector Quantization." *arXiv preprint arXiv:2406.11838* (2024).
> > >
> > > [5] Kenton, Jacob Devlin Ming-Wei Chang,  and Lee Kristina Toutanova. "Bert: Pre-training of deep bidirectional  transformers for language understanding." *Proceedings of naacL-HLT*. Vol. 1. 2019.
> > >
> > > [6] Chang, Huiwen, et al. "Maskgit: Masked generative image transformer." *Proceedings of the IEEE/CVF Conference on Computer Vision and Pattern Recognition*. 2022.
> > >
> > > [7] Wu, Kevin E., et al. "Protein structure generation via folding diffusion." *Nature communications* 15.1 (2024): 1059.

---

> > > ### Author Response · Authors · 2024-11-23
> > > **About Follow Up Problems (More Discussions and Future Work)**
> > >
> > > Thank you for taking the time to review our responses! We have provided further demonstrations to address your follow-up questions and would like to check in to see if you have any additional comments or questions that we can address to further improve our work and potentially raise your assessment.
> > >
> > > We would greatly appreciate your reply!

---

> ### Comment · Area_Chair_3KEp · 2024-11-25
> **Please clarify your view and score.**
>
> Dear reviewer mmeW,
>
> Thank you for engaging in discussions with the authors. You state your concerns have been addressed. This usually implies that you think the paper could be above the bar for acceptance, but I see that your score is still at 5. Could you help clarify what your current score is and your recommendation for accepting this paper or not, accompanied by a motivation?
>
> Many thanks,
>
> AC

---

### Official Review · Reviewer_UnBf · 2024-11-02

**Soundness:** 4
**Presentation:** 3
**Contribution:** 3
**Rating:** 6
**Confidence:** 4

**Summary:**

This paper proposes a method for peptide sequence and structure generation for peptide design with target protein. The methodology includes three parts: the hot-spot residue generation, the autoregressive generation of fragments based on the hot-spot residues to complete the chain, the correction stage that refine the peptide structure. Experiments have been performed on the peptide design and peptide scaffold generation tasks, and compared with baseline models like RFdiffusion, Pepflow, and demonstrate good performance in novelty and diversity, but not as good in the validity or quality of the generated structures.

**Strengths:**

- The methodology is novel, which divide-and-conquer the peptide design task into three stages: hot-spot residue generation, fragment completion and correction.
- Utilizing the information of hot-spot residues helps to improve the generation quality
- Experiments have been performed on the curated dataset and benchmarks, with comprehensive metrics, and comparison with relevant models.
- The method achieves the highest novelty and diversity compared with different baselines.

**Weaknesses:**

- The quality of the generated peptide is not good, on the metrics such as validity and stability the method is significantly worse.
- In the first stage, each hotspot generation is independent and do not consider other generated hotspot, or a joint generation. And in the second stage, the completion of each fragment only depends on the target protein and the generated residues of the current fragment. Not taking other hotspots or fragments into account in the two stages could make the model ignorant of important information for the geometry and validity.

**Questions:**

- In the correction stage, are the terms in J_bb only non-zero in those residues that stay at the connection of two fragments? Because other residues in the middle of the fragments should be complying with the consistency constraint from construction. The authors should clarify this.
- How does the correction method proposed in this work (using models directly from the first and second stages) compare with the correction by some empirical methods (like some energy function)?

---

> ### Author Response · Authors · 2024-11-19
> **Response to Reviewer UnBF**
>
> Thank you for your insightful and constructive comments as well as your appreciation of our work that **divide-and-conquer the peptide design task and utilize hot spot residue information**. Below are some clarifications and answers to your questions. If our response does not fully address your concerns, please post additional questions; we will be happy to discuss further.
>
> ## Q1: Not taking other hotspots or fragments into account in the two stages could make the model ignorant of important information for the geometry and validity.
>
> Thank you for raising this concern. We have clarified the distinction between independent and dependent sampling in our method. At the first stage, hotspots are sampled independently, as our distribution is defined based on residue distributions around the target. This approach makes independent sampling more efficient. In the extension stage, however, generation is conditioned on all existing fragments, including both the target and other residues from the current fragments. As noted in line 279, we explicitly clarify that ***“E represents the surrounding residues, including the target T and other residues in the currently generated fragments.”***
>
> ## Q2: In the correction stage, are the terms in $J_{\text{bb}}$ only non-zero in those residues that stay at the connection of two fragments? Because other residues in the middle of the fragments should be complying with the consistency constraint from construction. The authors should clarify this.
>
> In the first optimization step, if we only consider the $J_{\text{bb}}$​ term, the residues in the middle of the fragments, which are constructed through dihedral transformation, remain unchanged, while the positions and orientations of the connecting residues are modified, as you noted. However, since we jointly optimize both angles and positions for all residues using $J_{\text{bb}}$ and $J_{\text{angle}}$, updates to the connecting residues can influence their neighboring residues. Additionally, $J_{\text{angle}}$​ directly affects the dihedral angles of the middle residues. As a result, middle residues may also be adjusted in subsequent optimization steps. Overall, the correction stage refines the entire peptide structure. This approach aligns with the physical rationale: when fragments are connected to form a longer peptide, every residue may undergo structural and sequence-level adjustments to ensure consistency and validity.
>
> ## Q3: How does the correction method proposed in this work (using models directly from the first and second stages) compare with the correction by some empirical methods (like some energy function)?
>
>
> Our correction stage shares similarities with empirical methods, as both use objective functions to iteratively refine protein structures. However, there are key differences in the design and focus of the methods. Traditional approaches, such as Rosetta [1], FoldX [2], OpenMM [3], Madra X [4] energy functions, rely on physically inspired force fields and finely tuned parameters to optimize both backbone and side-chain structures. In contrast, our method specifically focuses on updating peptide backbone structures and sequences by leveraging gradients from backbone and angle-related terms. This targeted design makes our approach particularly effective for tasks involving backbone angle optimization and sequence updates, though less comprehensive in addressing side-chain adjustments.
>
> To provide a quantitative comparison, we evaluated the performance of our method alongside Rosetta's relax function (backbone movement on):
>
> | Method          	| Valid (%) |
> |---------------------|-----------|
> | PepHAR (K=3)    	| 59.31 	|
> | PepHAR w/o Correction | 53.66   |
> | PepHAR w/o Correction + Relax  	| 64.90 	|
>
> While our correction method does not achieve the highest performance compared to Rosetta relax, it is clearly more effective than omitting correction altogether. We believe that integrating more finely designed energy functions or adopting aspects of traditional empirical methods could further enhance validity rates. We will include additional discussion on the potential for combining our approach with more sophisticated traditional energy-based refinements.
>
> ## References:
>
> [1] Alford, Rebecca F., et al. "The Rosetta all-atom energy function for macromolecular modeling and design." Journal of chemical theory and computation 13.6 (2017): 3031-3048.
>
> [2] Delgado, Javier, et al. "FoldX 5.0: working with RNA, small molecules and a new graphical interface." Bioinformatics 35.20 (2019): 4168-4169.
>
> [3] Eastman, Peter, et al. "OpenMM 7: Rapid development of high performance algorithms for molecular dynamics." PLoS computational biology 13.7 (2017): e1005659.
>
> [4] Orlando, Gabriele, et al. "Integrating physics in deep learning algorithms: a force field as a PyTorch module." Bioinformatics 40.4 (2024): btae160.

---

> > ### Comment · Reviewer_UnBf · 2024-11-26
> >
> > Thank you to the authors for the response! All my questions have been well addressed. I acknowledge the authors' openness and comprehensive investigations with ablation studies. I will maintain my score.

---

> > > ### Author Response · Authors · 2024-11-26
> > >
> > > Thank you for your kind words and thoughtful evaluation—we greatly value your support and encouragement!

---

### Official Review · Reviewer_k8mv · 2024-11-03

**Soundness:** 4
**Presentation:** 4
**Contribution:** 3
**Rating:** 6
**Confidence:** 4

**Summary:**

This paper introduces PepHAR (Hotspot-driven Autoregressive peptide design), a novel three-stage approach for designing peptides that can bind to specific target proteins. The method acknowledges the critical role of "hot spot" residues in peptide-protein binding and addresses key challenges in computational peptide design. The three stages consist of: (1) a founding stage that uses an energy-based density model to identify and sample key hot spot residues, (2) an extension stage that autoregressively builds peptide fragments from these hot spots using a dihedral angle-based approach ensuring proper geometry, and (3) a correction stage that refines the complete structure through optimization. The authors also introduce a new "scaffold generation" task that better reflects real-world peptide drug development scenarios. When compared to state-of-the-art baselines (RFDiffusion, ProteinGenerator, PepFlow), PepHAR demonstrates superior performance across multiple metrics including structural validity, geometric accuracy, binding site accuracy, and novelty/diversity of generated peptides, while maintaining competitive stability and binding affinity.

**Strengths:**

- Novel Problem Decomposition: The work introduces a clever way to break down peptide design into a three-stage process based on biological understanding. By focusing on hot spot residues first, then extending and refining, it makes the problem more tractable while maintaining biological relevance.


- Strong Geometric Constraints: The method explicitly handles peptide bond geometry through dihedral angles. Uses von Mises distribution (and others) to model angle distributions properly.

- Novel Energy-Based Density Model: Introduces an innovative energy-based model to identify hot spot residues. Uses noise contrastive estimation for effective training without needing explicit normalization. Employs Langevin dynamics sampling to efficiently explore the residue distribution space.

- Autoregressive Fragment Extension: Enables bidirectional growth from hot spots through Left/Right operations


- Hybrid Optimization Framework: Creates a novel joint optimization objective combining backbone and dihedral constraints. Introduces an iterative refinement strategy that can handle multiple fragments simultaneously

**Weaknesses:**

Overall, this work is well written with clear logic and reasonable benchmarks.

1. Many recent works of peptide binder design use AlphaFold2 for scoring (e.g., pAE or iptm). I would like to see the evaluation of designed peptides with such metrics.

2. Regarding Novelty score, can author also show separate metrics for it in supplementary as well: TM-score and sequence identity.

3. For all examples provided, I will suggest authors to also provide the peptide sequence for reference for better understanding.

4. To enhance further the evaluation and discussion, I would suggest authors to include following works for discussion and potentially use them for benchmarking as well.

[1] EvoBind: https://www.biorxiv.org/content/10.1101/2022.07.23.501214v1 (Discussed, but not benchmarked?)

[2] EvoBind2: https://www.biorxiv.org/content/10.1101/2024.06.20.599739v2

[3] PepMLM: https://arxiv.org/abs/2310.03842

[4] EvoPlay: https://www.nature.com/articles/s42256-023-00691-9

[5] AlphaProteo: https://arxiv.org/abs/2409.08022 (very recent)

[6] MASIF: https://www.nature.com/articles/s41592-019-0666-6

**Questions:**

Major:
1. For different sampling K values, does the binder length remain constant for the same target protein?

2. RMSD might be somewhat misleading since peptides can still bind effectively with very different geometries, which is actually desirable - valid binders that show different binding modes from native ones.

3. Regarding Novelty, especially for sequences, what is the source of this novelty? Does it primarily come from scaffold sites or contact sites? If changes are mainly in scaffold sites compared to native ones, this metric might be misleading and trivial for a designed binder.

4. In the Dataset section, the binding pocket is defined using a 10Å radius, while for BSR in the supplementary material, binding sites are defined using a 6Å radius. Why are different values used?

Minor:
1. Authors should specify that they are performing peptide binder design, not peptide design, as these are distinctly different tasks.

2. In the Dataset section (Lines 276-377), please clarify the distinction between the 158 examples and the 8,207 additional examples.

---

> ### Author Response · Authors · 2024-11-19
> **Response to Reviewer k8mv, Part 1**
>
> Thank you for your insightful and constructive comments as well as your appreciation of our work in **introducing novel problem decomposition and proposing novel energy-based density model and autoregressive fragment extension**. Below are some clarifications and answers to your questions. If our response does not fully address your concerns, please post additional questions; we will be happy to discuss further.
>
> ## Q1: Many recent works of peptide binder design use AlphaFold2 for scoring (e.g., pAE or iptm). I would like to see the evaluation of designed peptides with such metrics.
>
> In recent works, AlphaFold2 is often used to predict peptide-receptor complex structures and score them using metrics like pLDDT, pAE, and ipTM [2,3,5]. Since there is no gold-standard metric, we propose an empirical evaluation using AlphaFold2 Multimer's ipTM score. This score measures the relative positioning accuracy of the complex’s subunits, with ipTM scores below 0.6 suggesting a failed prediction [7].
>
> We concatenated each generated peptide with the full receptor sequence (not just the binding pocket), and used AlphaFold2 Multimer to predict the complex structures. We then computed the proportion of complexes with ipTM scores above 0.6, indicating successful predictions. This metric, referred to as the "success rate," has been added to Tables 1 and 2 of the revised draft. Below, we also present the success rates for reference. As shown in tables, PepHAR outperforms PepGLAD (latent diffusion) and achieves comparable results to other baselines.
>
> ***Peptide Binder Design***
> | Method         	| Success % ↑ |
> |--------------------|-------------|
> | RFDiffusion    	| 25.38   	|
> | ProteinGenerator   | 24.43   	|
> | PepFlow        	| 27.96   	|
> | PepGLAD        	| 14.05   	|
> | PepHAR ($K=1$) 	| 23.00   	|
> | PepHAR ($K=2$) 	| 22.85   	|
> | PepHAR ($K=3$) 	| 25.54   	|
>
> ***Peptide Scaffold Generation***
> | Method              	| Success % ↑ |
> |-------------------------|-------------|
> | RFDiffusion ($K=3$) 	| 23.15   	|
> | ProteinGenerator ($K=3$)| 20.42   	|
> | PepFlow ($K=3$)     	| 24.54   	|
> | PepGLAD ($K=3$)     	| 14.85   	|
> | PepHAR ($K=1$)      	| 20.43   	|
> | PepHAR ($K=2$)      	| 22.09   	|
> | PepHAR ($K=3$)      	| 21.45   	|
>
>
> We would like to emphasize that scoring using structure prediction models has certain limitations. For example, we occasionally observe that AlphaFold2 fails to position peptides accurately within the binding site, resulting in low prediction confidence. Similarly, we find that ESMFold [9] tends to assign high confidence scores to single-chain peptides but consistently assigns low confidence scores to peptides in complex structures. Therefore, we believe that evaluating binder design from multiple perspectives is essential to identify the most promising candidates for wet-lab experiments [6].
>
> ## Q2: Regarding Novelty score, can author also show separate metrics for it in supplementary as well: TM-score and sequence identity.
>
> We have provided separate TM-score and sequence identity (AAR) metrics in Appendix G.1 Table 4, and we summarize the results here as well. The TM-score indicates that PepHAR tends to generate peptides that are less similar to native peptides, contributing to higher diversity and novelty. For AAR, PepHAR achieves sequence recovery comparable to RFDiffusion and ProteinGenerator and performs better than PepGLAD. However, relying solely on AAR is not sufficient to evaluate generative models, as it provides limited insight into the ability to capture the broad and diverse distribution of binding peptides. This limitation has also been discussed in the appendix of PepGLAD [10].
>
> ***Peptide Binder Design***
>
> | Method       	| TM   | AAR   |
> | ---------------- | ---- | ----- |
> | RFDiffusion  	| 0.44 | 40.14 |
> | ProteinGenerator | 0.43 | 45.82 |
> | PepFlow      	| 0.38 | 51.25 |
> | PepGLAD      	| 0.29 | 20.59 |
> | PepHAR (K=1) 	| 0.33 | 32.32 |
> | PepHAR (K=2) 	| 0.32 | 39.91 |
> | PepHAR (K=3) 	| 0.34 | 34.36 |
>
> ***Peptide Scaffold Generation***
> | Method             	| TM   | AAR   |
> | ---------------------- | ---- | ----- |
> | RFDiffusion (K=3)  	| 0.46 | 31.14 |
> | ProteinGenerator (K=3) | 0.48 | 32.05 |
> | PepGLAD (K=3)     	| 0.30 | 21.48 |
> | PepFlow (K=3)      	| 0.37 | 51.90 |
> | PepHAR (K=1)       	| 0.33 | 32.90 |
> | PepHAR (K=2)       	| 0.35 | 35.06 |
> | PepHAR (K=3)       	| 0.38 | 35.34 |
>
> ## Q3: For all examples provided, I will suggest authors to also provide the peptide sequence for reference for better understanding.
>
> We have updated Figures 3 and 4 with generated peptide sequences and their sequence identities. Additionally, we corrected a misused PDB code in Figure 4.

---

> > ### Author Response · Authors · 2024-11-19
> > **Response to Reviewer k8mv, Part 2**
> >
> > ## Q4: To enhance further the evaluation and discussion, I would suggest authors to include following works for discussion and potentially use them for benchmarking as well.
> >
> > We believe that reviewing as many current peptide generation models as possible not only highlights our contribution but also benefits the broader research community. To this end, we have added more related works [1,2,3,4,5,6,10] in Appendix H and introduced a new baseline, PepGLAD [10], in our main results. We believe our evaluation covers representative approaches in structure-based peptide design, particularly those using generative models. Our reviews are as follows:
> >
> > ```
> > Recent advancements in computational peptide binder design have significantly benefited from deep learning. Bryant et al. introduced EvoBind, an in silico directed evolution platform that utilizes AlphaFold to design peptide binders targeting specific protein interfaces using only sequence information. Building on this, Li et al. developed EvoBind2, which extends EvoBind's capabilities by enabling the design of both linear and cyclic peptide binders of varying lengths solely from a protein target sequence, without requiring the specification of binding sites or binder sizes. Using MCTS simulations and reinforcement learning,Wang et al. engineers peptide binders and achieves comparable result to EvoBind. Chen et al. proposed PepMLM, a target-sequence-conditioned generator for linear peptide binders based on masked language modeling. By employing a novel masking strategy, PepMLM effectively reconstructs binder regions, achieving low perplexities and demonstrating efficacy in cellular models. Leveraging geometric convolutional neural networks to decipher interaction fingerprints from protein interaction surfaces, Gainza et al. developed MaSIF. Subsequently, BindCraft combined AF2-Multimer sampling with ProteinMPNN sequence design to optimize protein-protein interaction surfaces. Similar to our approach of defining key hotspot residues, AlphaProteo  also aims to generate high-affinity protein binders that specifically interact with designated residues on the target protein. PepGLAD  encodes full-atom peptide structures and sequences using an equivariant graph neural network, and applies latent diffusion on the peptide embedding space to explore peptide generation.
> > ```
> >
> > ## Q5: For different sampling K values, does the binder length remain constant for the same target protein?
> >
> > Yes. As shown in Algorithm 1, we add residues iteratively until reaching a predefined length, which we set equal to the native binder length, following prior works [10,11]. However, this length can be adjusted based on the application.
> >
> > ## Q6: RMSD might be somewhat misleading since peptides can still bind effectively with very different geometries, which is actually desirable - valid binders that show different binding modes from native ones.
> >
> > We admit that RMSD may not fully represent what makes a "good binder." However, as a metric for evaluating generative models, it indicates how well the model captures the distribution of peptides conditioned on the target, and it is widely used in antibody and peptide design [10,11,12]. In our benchmark, we also include complementary metrics, such as novelty and diversity, to evaluate variations in binding geometries and sequences of generated peptides. Additionally, we would like to emphasize that there is an inherent trade-off between designing peptides that closely resemble native ones and those that prioritize diversity. We will provide further clarification on these metrics in the manuscript to improve understanding and facilitate future use.
> >
> > ## Q7: Regarding Novelty, especially for sequences, what is the source of this novelty? Does it primarily come from scaffold sites or contact sites? If changes are mainly in scaffold sites compared to native ones, this metric might be misleading and trivial for a designed binder.
> >
> > In generative protein design, novelty refers to the ability of a model to generate proteins that differ from native ones in terms of structural or sequence-level similarity [13,14]. As mentioned in Q6, generating diverse peptides can help explore alternative binding modes or contact sites, potentially leading to improved functionalities. To calculate novelty, we combine structural and sequence-level similarities into a quantitative metric that reflects the proportion of generated peptides distinct from the native ones.

---

> > > ### Author Response · Authors · 2024-11-19
> > > **Response to Reviewer k8mv, Part 3**
> > >
> > > ## Q8: In the Dataset section, the binding pocket is defined using a 10Å radius, while for BSR in the supplementary material, binding sites are defined using a 6Å radius. Why are different values used?
> > >
> > > We follow the approach used in PepFlow [11], which defines the binding pocket with a 10Å radius and the binding site with a 6Å radius. The binding pocket serves as the input condition for generating peptides, while the binding site is a smaller region within the binding pocket that potentially interacts closely with the peptide. These represent different sizes with distinct purposes.
> > >
> > > ## Q9: Authors should specify that they are performing peptide binder design, not peptide design, as these are distinctly different tasks.
> > >
> > > We are indeed performing receptor-conditioned peptide binder design. For clarity, we have updated the terminology from “peptide design” to “peptide binder design” in the main text.
> > >
> > > ## Q10: In the Dataset section (Lines 276-377), please clarify the distinction between the 158 examples and the 8,207 additional examples.
> > >
> > > The 8,207 examples are used for training and validation, while the 158 examples are reserved for testing. To prevent data leakage, we ensure that the test set examples are from different clusters than the training and validation sets, as determined using MMseqs2.
> > >
> > > ## References:
> > > [1] Bryant, P., & Elofsson, A. (2022). EvoBind: in silico directed evolution of peptide binders with AlphaFold. bioRxiv, 2022-07.
> > >
> > > [2] Li, Q., Vlachos, E. N., & Bryant, P. (2024). Design of linear and  cyclic peptide binders of different lengths only from a protein target  sequence. bioRxiv, 2024-06.
> > >
> > > [3] Chen, T., et al. "PepMLM: Target Sequence-Conditioned Generation of  Therapeutic Peptide Binders via Span Masked Language Modeling." 2023,
> > >
> > > [4] Wang, Yi, et al. "Self-play reinforcement learning guides protein engineering." Nature Machine Intelligence 5.8 (2023): 845-860.
> > >
> > > [5] Gainza, Pablo, et al. "Deciphering interaction fingerprints from protein molecular surfaces using geometric deep learning." Nature Methods 17.2 (2020): 184-192.
> > >
> > > [6] Zambaldi, Vinicius, et al. "De novo design of high-affinity protein binders with AlphaProteo." arXiv preprint arXiv:2409.08022 (2024).
> > >
> > > [7] https://www.ebi.ac.uk/training/online/courses/alphafold/inputs-and-outputs/evaluating-alphafolds-predicted-structures-using-confidence-scores/confidence-scores-in-alphafold-multimer/
> > >
> > > [8] Evans, Richard, et al. "Protein complex prediction with AlphaFold-Multimer." biorxiv (2021): 2021-10.
> > >
> > > [9] Lin, Zeming, et al. "Evolutionary-scale prediction of atomic-level protein structure with a language model." Science 379.6637 (2023): 1123-1130.
> > >
> > > [10] Kong, Xiangzhe, et al. "Full-atom peptide design with geometric latent diffusion." arXiv preprint arXiv:2402.13555 (2024).
> > >
> > > [11] Li, Jiahan, et al. "Full-Atom Peptide Design based on Multi-modal Flow Matching." arXiv preprint arXiv:2406.00735 (2024).
> > >
> > > [12] Luo, Shitong, et al. "Antigen-specific antibody design and optimization with diffusion-based generative models for protein structures." Advances in Neural Information Processing Systems 35 (2022): 9754-9767.
> > >
> > > [13] Yim, Jason, et al. "SE (3) diffusion model with application to protein backbone generation." International Conference on Machine Learning. PMLR, 2023.
> > >
> > > [14]  Bose, Joey, et al. "SE (3)-Stochastic Flow Matching for Protein Backbone Generation." The Twelfth International Conference on Learning Representations. 2023.

---

> > > > ### Comment · Reviewer_k8mv · 2024-11-23
> > > >
> > > > Thank you, authors, for addressing my previous concerns. I have one remaining question regarding potential side chain clashes. Could you please clarify if this issue occurs in your model? If so, what (potential) strategies have you implemented to resolve it, and what is the current rate of these clashes?

---

> > > > > ### Author Response · Authors · 2024-11-23
> > > > > **Follow Up Response: Regaring Poteintial Side Chain Clashes**
> > > > >
> > > > > Thank you for your follow-up question regarding potential side-chain clashes. We would like to clarify that our model generates only peptide backbone structures and sequences, following a widely adopted approach used in methods like RFDiffusion and ProteinGenerator. Our model does not directly generate side chains, so side-chain clashes are not encountered during backbone generation.
> > > > >
> > > > > In our evaluations, such as energy calculations, side chains are added and optimized using Rosetta’s energy functions based on the generated sequences. This approach is applied consistently across all baseline methods.
> > > > >
> > > > > To summarize, as our model does not directly generate side chains, this issue does not arise in our current framework. However, we recognize the importance of addressing side-chain clashes and will consider this challenge in future work when generating full-atom structures.

---

> > > > > > ### Comment · Reviewer_k8mv · 2024-11-23
> > > > > >
> > > > > > Thank you! I believe all my questions have been addressed. This is a good quality of work, and I’ll maintain my score of 6.

---

> > > > > > > ### Author Response · Authors · 2024-11-26
> > > > > > >
> > > > > > > Thank you for your positive feedback and support—we truly appreciate your recognition of our work!

---

### Author Response · Authors · 2024-11-19
**Global Response to All Reviewers**

We thank all reviewers for their kind and constructive suggestions, which have been invaluable in improving our work. Below, we summarize the key suggestions and our responses, along with the updates made to the draft:

1. **Baseline Models (4Qbp)**: We included an additional baseline, PepGLAD, in both the peptide binder design and peptide scaffold generation tasks, as shown in Tables 1 and 2.

2. **AF2 Metrics (k8mv)**: We introduced new AF2-based metrics, including success rate based on the ipTM score, which are now included in Tables 1 and 2.

3. **Hotspot Exploration (4Qbp)**: We explored different numbers of hotspots (K values) for all baseline models in the scaffold generation task. The results are presented in Appendix Section G.3, Table 7.

4. **Example Details (k8mv, mmeW)** We added sequence recovery rates and additional examples to Figures 4 and 5.

5. **Additional Results (k8mv, mmeW, Gac7)**:  We update additional result regarding Cumulative Errors (Appendix G.2), Case Study (Appendix G.4), Seconadary Structure Analysis (Appendix G.5), and review more recent peptide design works (Appendix H).

5. **Minor Revisions (k8mv, mmeW)**: Several minor issues were corrected and are highlighted in blue in the revised draft.

---

### Comment · Area_Chair_3KEp · 2024-11-25
**Last day for reviewers to ask questions to the authors!**

Dear reviewers,

Tomorrow (Nov 26) is the last day for asking questions to the authors. With this in mind, please read the rebuttal provided by the authors and their latest comments, as well as the other reviews. If you have not already done so, please explicitly acknowledge that you have read the rebuttal and reviews, provide your updated view and score _accompanied by a motivation_, and raise any outstanding questions for the authors.

**Timeline**: As a reminder, the review timeline is as follows:
- November 26: Last day for reviewers to ask questions to authors.
- November 27: Last day for authors to respond to reviewers.
- November 28 - December 10: Reviewer and area chair discussion phase.

Thank you again for your hard work,

Your AC

---

### Meta-Review · Area_Chair_3KEp · 2024-12-20

**Metareview:**

All but one reviewer has recommended to accept this submission. Reviewer mmeW has scored the submission with a 5, but has also explicitly indicated that the rebuttal addressed their concerns. Unfortunately the reviewer has not engaged further, but given the aforementioned statement I will consider their concerns addressed and that they are in favor of accepting this submission. Altogether, this leads me to recommend accepting this paper.

**Additional Comments On Reviewer Discussion:**

During the rebuttal the authors have included additional results, and added clarifications based on the reviewer questions. This has addressed most of the concerns by the reviewers.

---

### Decision · Program_Chairs · 2025-01-22

Accept (Poster)